# Novel Phenothiazine-Bridged Porphyrin-(Hetero)aryl dyads: Synthesis, Optical Properties, In Vitro Cytotoxicity and Staining of Human Ovarian Tumor Cell Lines

**DOI:** 10.3390/ijms21093178

**Published:** 2020-04-30

**Authors:** Eva Molnar, Emese Gal, Luiza Gaina, Castelia Cristea, Eva Fischer-Fodor, Maria Perde-Schrepler, Patriciu Achimas-Cadariu, Monica Focsan, Luminita Silaghi-Dumitrescu

**Affiliations:** 1The Research Center on Fundamental and Applied Heterochemistry, Department of Chemistry, Faculty of Chemistry and Chemical Engineering, Babeş-Bolyai University, 11 Arany Janos street, RO-400028 Cluj-Napoca, Romania; molnar_evike@yahoo.com (E.M.); emese@chem.ubbcluj.ro (E.G.); gluiza@chem.ubbcluj.ro (L.G.); lusi@chem.ubbcluj.ro (L.S.-D.); 2Department of Radiobiology and Tumor Biology, Institute of Oncology “Prof.Dr. Ion Chiricuta”, 34-36 Republicii street, RO-400015 Cluj-Napoca, Romania; efischerfodor@yahoo.com (E.F.-F.); pmariaida@yahoo.com (M.P.-S.); 3Medfuture Research Center for Advanced Medicine, University of Medicine and Pharmacy Iuliu Hatieganu, 8 Babes street, RO-400012 Cluj-Napoca, Romania; 4Department of Surgery, Institute of Oncology “Prof.Dr. Ion Chiricuta”, 34-36 Republicii street, 400015 Cluj-Napoca, Romania; patrick.achimas@hotmail.com; 5Department of Surgery and Gynecological Oncology, University of Medicine and Pharmacy Iuliu Hatieganu, 23 Marinescu street, 400337 Cluj-Napoca, Romania; 6Institute for Interdisciplinary Experimental Research in Bionanoscience, Nanobiophotonics Laboratory, Babeş-Bolyai University, 42 Laurian street, 400271 Cluj-Napoca, Romania; monica.iosin@phys.ubbcluj.ro

**Keywords:** *meso*-aryl-porphyrin, metal-porphyrin, phenothiazine, carbazole, UV–Vis spectroscopy, cytotoxicity, fluorescence imaging, TPE-FLIM

## Abstract

We report here the synthetic procedure applied for the preparation of new AB_3_-type and *trans*-A_2_B_2_ type *meso*-halogenophenothiazinyl-phenyl-porphyrin derivatives, their metal core complexation and their peripheral modification using Suzuki–Miyaura cross coupling reactions with various (hetero)aryl (phenothiazinyl, 7-formyl-phenothiazinyl, (9-carbazolyl)-phenyl and 4-formyl-phenyl, phenyl) boronic acid derivatives. The *meso*-phenothiazinyl-phenyl-porphyrin (MPP) dyes family was thus extended by a series of novel phenothiazine-bridged porphyrin-(hetero)aryl dyads characterized by UV–Vis absorption/emission properties typical to the porphyrin chromophore, slightly modulated by increasing the size of peripheral substituents. Three phenothiazine-bridged porphyrin-heteroaryl dyads with fluorescence emission above 655 nm were selected as fluorophores in red spectral region for applications in cellular staining of human ovarian tumors. In vitro experiments of cell metabolic activity displayed a moderate toxicity on human ovarian tumor cell lines (OVCAR-3, cisplatin-sensitive A2780 and cisplatin-resistant A2780cis respectively). Visualization of the stained living cells was performed both by fluorescence microscopy imaging and by fluorescence lifetime imaging under two photon excitation (TPE-FLIM), confirming their cellular uptake and the capability of staining the cell nucleus.

## 1. Introduction

The multiple functionalization possibilities of the highly stable planar macrocyclic porphin core combined with the huge progress recorded in the development of methodologies for synthetic organic chemistry mostly concurred with the expansion of recent literature data documenting new porphyrin derivatives with tailored chemical and optical properties. Introduction of appropriate donor/acceptor substituents and/or metal complexation contributed to the modification of structural factors, such as molecular symmetry, size and degree of conjugation of the π-electronic system, emphasizing a high impact on the optical performances of the porphyrin-based materials. Among the various heteroaromatic units attached to the porphine chromophore aiming the modulation of its photophysical properties, the introduction of an electron rich phenothiazine unit gave fruitful returns; e.g., fluorescent porphyrins emitting intense red light with high fluorescent quantum yields [1] “push−pull” porphyrin dyes for photon to energy conversion systems [2,3,4,5] as well as potential biologically active compounds [6]. The syntheses, optical properties and electrochemical behaviors of *meso*-phenothiazinyl-phenyl porphyrin (MPP) dyes were first reported by our group [7] followed by the description of their metal complexation [8].

The Suzuki–Miyaura cross-coupling between organoboron derivatives and organic halides in the presence of a palladium catalyst and a base became a powerful methodology for carbon–carbon bond formation [9]. This palladium-mediated cross-coupling reaction heavily benefits from the availability of: organoboron reagents (which are inert to water and oxygen, and characterized by thermal stability and low toxicity [10]); tetra-coordinated palladium-phosphine complex catalysts, such as Pd(PPh_3_)_4_, Pd(PPh_3_)_2_Cl_2_, Pd(dppf)Cl_2_ or Pd_2_(dba)_3_ [11]; and different bases, such as Cs_2_CO_3_ [12], K_2_CO_3_ [13], KO*t*Bu [14], K_3_PO_4_ [15] or NEt_3_ [16].

A vast array of methodology options became accessible taking into consideration that both phenothiazine and porphyrin derivatives can be used as coupling partners in Suzuki–Miyaura cross coupling reactions either as halogeno or arylboronic derivatives. Phenothiazinylboronic acid derivatives were successfully synthesized from brominated phenothiazines either by bromine–lithium exchange followed by trapping with trialkylborate or by palladium-catalyzed borylation with tetramethyl dioxoborolane [17], while various (hetero)aryl substituents were attached to the phenothiazine core by Suzuki cross-coupling when starting with commercial chlorophenothiazine [18] or readily accessible 3,7-halogeno-phenothiazine derivatives [19,20,21]. The coupling of A_2_B_2_-type *meso*-bromoporphyrin with arylboronic acid derivatives was performed under mild conditions (aqueous Na_2_CO_3_ as a base in DMF) [22]. Starting from *meso*-borylporphyrin and *meso*-bromoporhyrin, [23] a *meso*-*meso*-linked porphyrin system was obtained using catalytic amounts of Pd(PPh_3_)_4,_ Cs_2_CO_3_ in DMF/toluene solution. A combination of Cs_2_CO_3_ and a Pd(0)-complex was employed in the cross coupling of iodoferrocene and 5,15-diphenyl-10-borylporphyrin or other bulky boronic acids [24]. (e.g., polycyclic aromatic hydrocarbons [25].) 

In this work, the MPP dyes family was expanded through the preparation of phenothiazine-bridged porphyrin-(hetero)aryl dyads. The described synthetic methodology is based on Suzuki–Miyaura cross coupling reaction between AB_3_ or *trans*-A_2_B_2_ type *meso*-bromophenothiazinyl-phenyl-porphyrines (generally labeled BrMPP) with (hetero)aryl boronic acid derivatives (phenothiazinyl, substituted phenyl). Novel A_4_-type phenyl-bridged porphyrin-phenothiazine dyads were obtained by Suzuki–Miyaura cross coupling between *meso-tetrakis*(4-bromo-phenyl)porphyrin (BrTPP) and phenothiazinyl-boronic acid derivatives. The optical UV–Vis absorption/emission properties of the newly synthesized compounds were investigated in our pursuit for π-expanded porphyrins characterized by a bathochromic shift of the typical absorption/emission maxima. The effect of metal core substitution on the UV–Vis optical response of AB_3_ or *trans*-A_2_B_2_ type halogenoMPP was also examined. Aiming the distinct visualization of human ovarian tumors, three phenothiazine-bridged porphyrin-heteroaryl dyads were tested as fluorophores in red spectral region for cellular staining of human ovarian tumors; following this goal their cytotoxicity was studied by means of colorimetric assays, while their ability to stain human ovarian tumor cell lines was proved by both fluorescence microscopy imaging and non-invasive two photon excited-fluorescence lifetime imaging microscopy (TPE-FLIM) technique.

## 2. Results

### 2.1. Synthesis of Novel MPP Dyes

#### 2.1.1. *Meso*-(Halogenophenothiazinyl-Phenyl)Porphyrins 

*Meso*-Halogenophenothiazinyl-phenylporphyrin dyes (halogen-MPP) of AB_3_-type (halogen-MPP **2** containing one peripheral halogenophenothiazine unit) and *trans*-A_2_B_2_ type (halogen-MPP **3** containing two peripheral halogenophenothiazine units) presented in Scheme 1 were prepared by mixed condensation of pyrrole with benzaldehyde and 7-bromo-10-methyl-phenothiazin-3-carbaldehyde **1a** or 8-chloro-10-methyl-phenothiazin-3-carbaldehyde **1b** respectively, based on our previously reported procedure, which affords MPP dyes containing a variable number of phenothiazinyl units attached to the peripheral *meso* positions of the central porphyrin core [7]. The presence of the halogen atom on the phenothiazine carbaldehyde substrate did not impede the outcomes of the macrocycle synthesis (yields 10–18%) and the separation of **2** and **3** obtained in 3:2 weight ratio was done by column chromatography. The starting bromophenothiazine carbaldehyde **1a** was prepared from phenothiazine substrate by successive N-methylation; Vilsmeier formylation; and finally, bromination of the phenothiazine carbaldehyde intermediate [26], while **1b** was prepared starting with commercial 2-chlorophenothiazine which required protection by N-methylation before applying the similar Vilsmeier formylation procedure. 

#### 2.1.2. *Meso*-(Halogenophenothiazinyl-Phenyl)Porphyrin Metal Complexes

Metalation of halogenoMPP AB_3_ type **2a,b** and *trans*-A_2_B_2_ type **3a,b** with *d*-block elements possessing polarizable valence electrons capable of perturbing the π-electron system of the porphyrin core was performed in order to underscore the effects of metal core substitution on their reactivities and UV–Vis optical properties. The complexation of the central cavity of the halogen-MPP macrocycles **2**, **3** was accomplished by the treatment with the appropriate metal (II) acetate in *N*,*N*-dimethylformamide (DMF) solution [27]. In Scheme 2 are the metal-complexes obtained by incorporation of Zn(II), Pd(II) Ni(II) and Cu(II), metal ions respectively in halogen-MPP ligands containing one (**4**, **5**) or two peripheral phenothiazine units (**6**, **7**)

#### 2.1.3. Phenothiazine-Bridged Porphyrin-(Hetero)Aryl Dyads

The presence of the halogen substituent on the phenothiazine unit in AB_3_ type porphyrin **2a** offered the possibility for further extension of the aromatic π-electronic system by Suzuki–Miyaura cross coupling reaction with (hetero)aryl-(phenothiazinyl, 7-formyl-phenothiazinyl, 9-carbazolyl-phenyl, 4-formyl-phenyl or phenyl respectively) boronic acid derivatives, which generated the phenothiazine-bridged porphyrin-(hetero)aryl dyads **8**–**12**, whereas *trans*-A_2_B_2_ type porphyrin **3a** afforded the symmetric double phenothiazine-bridged porphyrin-(hetero)aryl dyads **13**–**15**, as depicted in Scheme 3. 

The cross coupling reaction was proven to be successfully performed when starting with BrMPP Zn(II) complex **4a** and phenothiazinyl-boronic acid or phenyl-boronic acid derivative respectively, generating the phenothiazine-bridged porphyrin-phenothiazine/phenyl dyads Zn(II) complexes **8a** and **11a**. 

#### 2.1.4. Phenylene-Bridged Porphyrin-(Hetero)Aryl Dyads

Under similar Suzuki–Miyaura cross coupling reaction conditions, *meso-tetrakis*(4-bromo-phenyl)porphyrin **16** and formyl-phenothiazine boronic ester or phenyl boronic acids (9-carbazolylphenyl-, 4-formyl-phenyl) afforded the novel A_4_ type phenylene-bridged porphyrin-(hetero)aryl dyads **17**–**19** (Scheme 4) in good yields (~60%). 

For each synthesized porphyrin, the structural assignments were based on the results of high-resolution mass spectrometry, ^1^H-NMR and ^13^C-NMR spectroscopy. The key signals generated by the porphyrin inner core protons were shifted upfield (δ = −2.7 ppm) in the free bases and disappeared from the ^1^H-NMR spectra of metal complexes **4**–**7**. The symmetrical structure of each of the *trans*-A_2_B_2_ type derivatives **3** and **13**–15 was unambiguously sustained by ^1^H-NMR spectra displaying four isochrone AB spin systems situated as low field signals (δ ≈ 8.9 ppm) for the eight protons situated in the *beta* positions of the pyrrole units encrypted in the macrocyclic structure (in Appendix A are presented the NMR spectra of **3a** (Appendix A), **5a** (Appendix A) and **13** (Appendix A) well as HRMS spectra of porphyrin ligand **3a** (Appendix A) and metallo complexes with Zn(II) (Appendix A), Pd(II) (Appendix A) and Cu(II) (Appendix A). 

### 2.2. Optical Properties of Novel MPP Dyes

#### 2.2.1. UV–Vis Absorption Spectra

The UV–Vis linear absorption optical properties of the newly synthesized porphyrin derivatives recorded in dichloromethane (DCM) solution confirmed the presence of both phenothiazine and porphyrin chromophore systems. A UV absorption band situated in the range 240–260 nm typical to the phenothiazine chromophore is accompanied by a strong near-UV Soret band with maxima situated in the range 418–427 nm followed by four low intensity Q bands located in the visible spectral region in the range 515–520 nm, characteristic of the free-base *etio*-type porphyrin chromophore [7]. The positions of the absorption maxima for the synthesized porphyrin derivatives are depicted in Table 1. All the compounds are characterized by a dark purple color as a consequence of the intense absorption in the Soret band region. By examining the molar absorptivity values listed in Table 1, it can be seen that halogen-MPP **2a** and A_4_ type *meso*-phenylene-bridged porphyrin-carbazole dyad **18** appear well fitted for dyes applications, while the presence of the phenothiazine bridge in porphyrin-(hetero)aryl dyads **8**–**15** does not promote a hyperchromic effect. 

As can be seen in Figure 1 depicting the UV–Vis absorption spectra of phenothiazine-bridged porphyrin phenothiazine dyads, a hyperchromic effect was recorded in the case of AB_3_ type dyad **9** sensitive to the presence of auxochromic formyl group, whereas in the case of symmetric substitution in *trans* A_2_B_2_ type dyad **13**, this hyperchromic effect appeared slightly diminished.

The characteristic UV–Vis absorption maxima recorded in solution for the prepared metal complexes of halogen-MPP are indicated in Table 2. As compared to the corresponding free base ligand, the electronic spectrum of each metal-complex displayed in the near UV range an intense Soret band without important shifting of its maxima position and in the visible region one or two Q bands, indicative of successful metal insertion in a square planar geometry [8].

#### 2.2.2. UV–Vis Fluorescence Emission Spectra

Upon irradiation around the λ_max_ corresponding to Soret band, in solution, phenothiazine-bridged porphyrin-(hetero)aryl dyads displayed fluorescence with noticeable Stokes shifts (above 8000 cm^−1^) and quantum yields between 0.08 and 0.19 when measured against TPP standard, as depicted in Table 3.

In Figure 2 are the emission spectra of phenothiazine-bridged porphyrin dyads containing phenothiazine **9** and **13** and carbazole **14** moieties, emphasizing the fluorophore properties of the symmetrical *trans* A_2_B_2_ type phenothiazine-bridged porphyrin-phenothiazine dyad **13**.

The presence of the phenothiazine units in dye scaffolds may be considered as responsible on one hand for the energy dissipation by typical butterfly vibrations, thereby inducing low fluorescence emission in solution, but on the other hand, for enlarging the intermolecular distance due to its bent structure, thereby preventing the intermolecular π–π stacking and subsequently stimulating fluorescence in the aggregated state [28]. In solution, the parent MPP dyes displayed fluorescence emissions typical to the porphyrin core in 660–670 nm range [7], while the extension of the pending phenothiazine arms of A_3_B type and A_2_B_2_ type MPP with additional peripheral phenothiazine or phenyl units produced fluorescence quenching in compounds **8**, and **11**. The presence of formyl auxochromic groups in peripheral positions appeared beneficial by imparting molecular polarization via an electron withdrawing effect, and thus fluorescence emissions were recorded for compounds **9**, **12**, **13** and **15**, as shown in Table 3. Phenothiazine-bridged porphyrin-carbazole dyads A_3_B type 10 and A_2_B_2_ type **14** displayed fluorescence emissions in similar range. The symmetrical A_4_ type phenylene-bridged porphyrin-(hetero)aryl dyads did not display fluorescence emission regardless the nature of the peripheral units: formyl-phenothiazine **17**, N-phenyl carbazole **18** and phenyl-carbaldehyde **19**.

### 2.3. Biological Activity Evaluation of Phenothiazine-Bridged Porphyrin-Phenothiazine Dyad Fluorescent Dyes 9, 13 and 15

Pursuing our goal of enabling the distinct visualization of human ovarian tumors, fluorescent dyes **9**, **13** and **15** (Table 3) displaying fluorescence emission in solution above 650 nm and containing the carbaldehyde functional units susceptible to passing on the affinity towards the bio-materials were examined in vitro in terms of their cytotoxicity and their capacity for staining living cells by subcellular localization in human ovarian tumor A2780, cisplatin-resistant ovarian adenocarcinoma cells A2780cis and OVCAR-3 cell lines. 

#### 2.3.1. MTT Colorimetric Assay

The cells’ viability was tested using the MTT colorimetric assay (dye 3-(4,5-dimethylthiazol-2-yl)-2,5-diphenyltetrazolium bromide) following a previously reported colorimetric protocol [29]. Separate serial dilutions of each dye **9**, **13** and **15** were employed in the treatment of ovarian adenocarcinoma cells lines under study. The modification in cell viability was observed by monitoring the concentration of the purple formazan formed after 24 h of incubation and reflected by the recorded optical density (OD) of the sample. The cells’ viability was expressed as percentage of survival in treated *versus* untreated (control) cell populations, and the results are shown in Table 4.

A close inspection of the data in Table 4 suggested that there were no significant differences between the cytotoxicity of the tested dyes **9**, **13** and **15**, each of them exhibiting a moderate toxicity on ovarian tumor cell lines in vitro (one-way analysis of variances, Bonferroni post-test, in the 95% confidence interval). From the viewpoint of the cell’s growth inhibition, A2780cis was the most resistant, while the folate receptor-rich OVCAR-3 cell line appeared the most susceptible to being affected. 

#### 2.3.2. Alamar Blue Assay

The metabolic activity of the ovarian adenocarcinoma cells lines under study was evaluated by the Alamar Blue fluorimetric test [30] in the presence of dyes **9**, **13** and **15** using the same eight serial dilutions ranging from about 250 to 2 µM employed in the MTT colorimetric assay (Table 4). After 24 h of incubation, the nonfluorescent Alamar Blue dye was reduced by cell metabolic activity to the pink fluorescent resofurin dye (excitation at 570 mn, emission detected at 585 nm). In Figure 3, the correlation between fluorescence intensity and concentration for the tested fluorescent dyes against ovarian adenocarcinoma cells A2780 is shown. (In the Appendix A there is a similar correlation for ovarian adenocarcinoma A2780cis (Appendix A) and OVCAR3 cells (Appendix A).

The data provided by the fluorescence assessment were used to build a linear regression of dose–response relationship for each tested compound, and to quantify the results with the mathematic parameter hillslope, as presented in Table 5.

The deepest drops in metabolic rate, represented by negative hillslopes (Table 5) have been recorded for cisplatin-sensitive A2780 (−655.4) and cisplatin-resistant A2780cis (−567.1) cell lines subsequent to the treatment with 13, while for the folate-rich OVCAR-3 cell line, the largest metabolic rate reduction was caused by **9** (−344.8).

#### 2.3.3. Cell Fluorescence Imaging

Fluorescence microscopy became an indispensable tool in single-molecule investigations, providing a high signal-to-noise ratio for visualization while still retaining the key features in the physiological context of native biological systems [31]. Fluorescent porphyrin derivatives characterized by large Stokes shift between absorption (~400 nm) and red/NIR emission were prospected in the fluorescence detection in various tissues or in vivo. In order to prevent the aggregation caused quenching effect due to π–π stacking aggregation favored by the planarity of the aromatic structure of the macrocycle, one chemical approach was to introduce various bulky groups as substituents of the porphyrin core [32]. Fluorescence imaging was carried out in order to investigate the internalization pathways of the selected fluorescent dyes **9**, **13** and **15** inside the tested ovarian adenocarcinoma cells. The typical chemical reactivity of the carbaldehyde unit was the selected structural feature meant for passing on the affinity towards the bio-materials. In phenothiazine-bridged porphyrin-phenothiazine dyads A_3_B type **9** and A_2_B_2_ type **13** the formyl group was attached to the bent phenothiazine unit(s) prone to prevent the intermolecular π–π stacking, and green fluorescence was displayed by these staining agents in the aggregated state. Dye **15** containing the formyl group attached to peripheral flat phenylene units exhibited aggregation-caused fluorescence quenching, which limited its application.

The fluorescent dye concentration applied for cells staining was 50 µM estimated and was well fitted for both fluorescence visualization and avoidance of cytotoxicity effects responsible for reducing the cell viability. Figure 4 illustrates the A2780cis cells in a typical Bright-field image (Figure 4a); their corresponding fluorescence imaging after the standard nuclei staining with DAPI (4′,6-diamidino-2-phenylindole) (Figure 4b); and fluorescence imaging after staining with fluorescent dye 13 (Figure 4c).

The merged fluorescence image presented in Figure 4d reveals the green fluorescent signal of dye **13** overlain with the blue fluorescent signal of DAPI distributed inside the cells. This result allows us to conclude that fluorescent dye **13** is able to accumulate both in the perinuclear and the nuclei region of the ovarian adenocarcinoma cells, being susceptible to crossing the nucleus membrane. 

In order to confirm the cellular uptake of dye **13**, we further invoked two photon excited-fluorescence lifetime imaging (TPE-FLIM), an ideal non-invasive technique for live-cell imaging, which offers multiple advantages over linear optical microscopy, including acquisition of fluorescence image with deep light penetration generated by NIR excitation, less photodamage of living organism and negligible background signal [33]. In recent literature, functional porphyrins were specified as potential fluorescent probes for TPE fluorescence imaging of various tumoral cells. Thus, 2-acetyl-6-dimethyl-aminonaphthalene–porphyrin dyads, characterized by intramolecular fluorescence resonance energy transfer process (FRET) from the aminonaphthalene donor to the porphyrin acceptor unit, were employed for TPE fluorescence imaging of lung cancer A549 cells under the irradiation with a 740 nm femtosecond laser. [34,35] Mesoporous silica nanocomposites containing the photosensitizer 5,10,15,20-*tetrakis*-(1-methyl-4-pyridinio)porphyrin-*tetra*-(*p*-toluenesulfonate) were employed for targeted TPE-FLIM of human breast carcinoma cell line MCF-7 and the human lung cancer A549 cell line [36]. A non-covalent complex of lipid-coated semiconductor CdSe/ZnS quantum dots with a partially hydrogenated porphyrin (chlorin e6) photosensitizer was imaged within plasma membrane and intracellular compartments of living HeLa cells [37]. In our experiments, the chosen wavelength of the excitation laser was 800 nm, ensuring that the combined energy of two photons spanned the gap between the ground state and first excited electronic state of the fluorescent dye **13**. Figure 5 illustrates the bright field image corresponding to A 2780 cells (Figure 5a) and the TPE-FLIM image where the contrast is based on the lifetime of the fluorophore **13** after staining A 2780 cells (Figure 5b).

The collected TPE-FLIM image presents the successful molecular binding of the dye **13** inside A2780 cells, giving the possibility of a distinct visualization of ovarian adenocarcinoma cells in NIR light.

## 3. Materials and Methods 

### 3.1. Spectral Measurements

The mass spectra were recorded on a HRMS spectrometer LTQ Orbitrap XL-Thermo-Scientific (Waltham, MA USA) using APCI or ESI ionization technique. NMR spectra were recorded at room temperature on Bruker Avance instruments (Bruker BioSpin GmbH, Rheinstetten, Germany) (^1^H/^13^C: 400 MHz/100 MHz or 600 MHz/150 MHz) in solution (deuterated solvents chloroform (CDCl_3_) or DMSO). UV–Vis spectra were recorded in CH_2_Cl_2_ with a Perkin Elmer Lambda 35 spectrophotometer (Waltham (HQ), MA, USA). Fluorescence spectra in solution were recorded in DCM with a Perkin Elmer PL 55 spectrophotometer (Waltham (HQ), MA, USA). The fluorescence quantum yield was calculated using tetraphenylporphyrin (TPP) standard Φ = 0.13% in DCM solution [38]. 

### 3.2. Reagents

Phenothiazine, 2-chlorophenothiazine and phenylboronic acid were purchased from Sigma Aldrich; *10-methyl-7-(4,4,5,5-tetramethyl-1,3,2-dioxaborolan-2-yl)-10H-phenothiazine-3-carbaldehyde* was prepared according to the procedure described in the literature [39].

*(4-(9H-carbazol-9-yl)phenyl)boronic acid* was prepared according to the procedure described in the literature [40].

*5,10,15,20-tetrakis(4-bromophenyl)-21H,23H-porphine* was prepared according to the procedure described in the literature [41].


*8-Cloro-3-formyl-10-methyl-phenothiazine 1b*


To a cooled solution (0 °C) containing DMF (10 mL) and POCl_3_ (12 mL), 2-chloro-10-methyl-10*H*-phenothiazine (30 g, 0.12 mol) dissolved in 1,2-dichloroethane (150 mL) was added, and the reaction mixture was heated at 85 °C for 6 h. The reaction mixture was poured into cold water (200 mL) and extracted with ethyl-acetate (2 × 200 mL). The extract was dried over anhydrous magnesium sulfate and the solvent was removed till dryness. The product was purified by column chromatography, using a silica gel solid support and toluene as eluent. Yellow powder, yield 48% (16.1g), ^1^H-NMR (400 MHz, CDCl_3_) δppm 3.44 (s, 3H, CH_3_), 6.81(d, 1H, ^4^*J* = 1.9 Hz), 6.88 (dd, 1H, ^3^*J* = 6.4 Hz, ^4^*J* = 1.9 Hz), 6.97 (d 1H, ^3^*J* = 6.4 Hz), 7.03 (d, 1H, ^3^*J* = 6.8 Hz), 7.60 (d, 1H, ^4^*J* = 1.8 Hz), 7.68 (dd, 1H, ^3^*J* = 6.8 Hz ^4^*J* = 1.8 Hz), 9.81 (s, 1H, CHO); ^13^C-NMR (100 MHz, CDCl_3_) δppm 35.9 (N-CH_3_), 114.1, 115.2, 121.0 (C_q_), 123.4 (C_q_), 123.8, 127.8, 127.9, 130.6, 131.5 (C_q_), 133.7 (C_q_), 145.3 (C_q_), 150.3 (C_q_), 190.0 (CHO).

#### 3.2.1. General Procedure for the Synthesis of AB_3_ and A_2_B_2_-Type *Meso*-Halogenophenothiazinyl-Porphyrins (2 and 3)

Propionic acid (100 mL) and acetic anhydride (3 mL) were stirred and heated at 110 °C for 1 h. After cooling at room temperature, benzaldehyde (*12mmol*), 10-methyl-10*H*-phenothiazinyl-carbaldehyde derivative (*12mmol*) and pyrrole (*24 mmol*) were added and the mixture was heated at 110 °C for 4 h. After completion, the obtained purple crystals were collected by filtration and washed with methanol to remove the traces of propionic acid. In the case in which the product did not precipitate, the solvent was removed by reduced pressure distillation and the residue was washed with methanol. The separation by column chromatography on silica gel gave the corresponding *meso*-halogenophenothiazinyl-porphyrins (2) and (3) respectively.

*5,10,15-triphenyl-20-(7-bromo-10-methyl-10H-phenothiazin-3-yl)-21,23H-porphyrin* 2a

Starting with bromo-phenothiazine carbaldehyde 1a (4 g, 0.012 mol), benzaldehyde (1.3 mL, 0.012 mol), pyrrole (1.7 mL, 0.024 mol), propionic acid (100 mL) and acetic anhydride (3 mL) were added, and after purification by column chromatography using dichloromethane/petrol ether (2:1) as eluent, a purple powder was obtained; yield 18% (0.9 g). ^1^H-NMR (400 MHz, CDCl_3_) δppm -2.73 (s, 2H, NH), 3.60 (s, 3H), 6.85 (d, 1H, *J* = 7.4 Hz), 7.15 (d, 1H, *J* = 7.4 Hz), 7.37–7.40 (m, 2H), 7.78–7.83 (m, 9H), 8.01–8.05 (m, 2H), 8.25 (d, 6H, *J* = 7.2 Hz), 8.86–8.89 (m, 6H), 8.92 (d, 2H, *J* = 4.5 Hz); ^13^C-NMR (100 MHz, CDCl_3_) ppm 35.6, 112.5, 115.0, 115.4, 118.7 (Cq), 120.2 (4Cq), 121.2 (Cq), 125.5 (Cq), 126.7 (9C), 127.7 (6C), 129.7, 130.3 (2Cq), 132.9, 134.0, 134.5 (8C), 136.8 (Cq), 142.1 (8Cq), 145.0 (3Cq). HRMS-APCI calcd. for: C_51_H_35_BrN_5_S [M+H]^+^ 828.1717; found: 828.1791.

*5,10,15-triphenyl-20-(2-chloro-10-methyl-10H-phenothiazin-7-yl)-21,23H-porphyrin* 2b

Starting with phenothiazine carbaldehyde 1b (4 g, 0.012 mol), benzaldehyde (1.3 mL, 0.012 mol), pyrrole (1.7 mL, 0.024 mol), propionic acid (100 mL) and acetic anhydride (3 mL), a mixture of 2b and 3b was obtained. After separation by column chromatography using chloroform/heptane (2:1) as eluent, 2b was obtained as a purple powder; yield 14% (0.4 g); ^1^H-NMR (600 MHz, CDCl_3_) δppm −2.76 (s, 2H, NH), 3.66 (s, 3H), 7.01(s, 1H), 7.04 (d, 1H, *J* = 7.9 Hz), 7.18 (d, 1H, *J* = 8.1 Hz), 7.21 (d, 1H, *J* = 7.9 Hz), 7.78–7.82 (m, 9H,), 8.02–8.06(m, 2H), 8.24 (d, 6H, *J* = 8.2 Hz), 8.87 (d, 6H, *J* = 4.2 Hz), 8.91 (d, 2H, *J* = 4.2 Hz); ^13^C-NMR (150 MHz, CDCl_3_) δ ppm 35.7, 112.6, 114.8, 118.7 (Cq), 120.21, 120.23 (Cq), 121.6 (Cq), 121.8 (Cq), 122.5, 126.7 (9C), 127.7 (6C), 127.9 (3Cq), 128.2 (Cq), 129.0 (Cq), 132.9, 133.7 (2Cq), 133.9, 134.5 (8C), 137.0 (2Cq), 142.1 (4Cq), 144.8 (2Cq), 147.0 (2Cq). HRMS-APCI calcd. for: C_51_H_35_ClN_5_S [M+H]^+^ 783.2223; found: 783.2206.

*5,15-diphenyl-10,20-bis(7-bromo-10-methyl-10H-phenothiazin-3-yl)-21,23H-porphyrin* 3a 

Purification by column chromatography using dichloromethane/petrol ether (2:1) as eluent gave a purple powder; yield 12% (0.8 g); ^1^H-NMR (400 MHz, CDCl_3_) δ ppm –2.75, (s, 2H, NH), 3.58 (s, 6H), 6.83 (d, 2H, *J* = 9.2 Hz), 7.14 (d, 2H, *J* = 8.4 Hz), 7.37–7.41 (m, 4H), 7.77–7.81 (m, 6H), 7.98–8.04 (m, 4H), 8.24 (d, 4H, *J* = 8.2 Hz), 8.87 (d, 4H, *J* = 4.2 Hz), 8.91 (d, 4H, *J* = 4.2 Hz); ^13^C-NMR (100 MHz, CDCl_3_) δppm 36.0 (2C), 112.5 (2C), 115.0 (2C), 115.4 (2C), 118.8 (2Cq), 120.2 (2Cq), 121.1 (Cq), 125.3 (Cq), 125.5 (2Cq), 126.7 (4C), 127.7 (2C), 128.2 (2Cq), 129.0 (2C), 129.7 (4C), 130.3 (4C), 132.9 (2C), 134.0 (2C), 134.5 (4C), 136.8 (4Cq), 142.1 (4Cq), 145.0 (4Cq), 145.1 (4Cq). HRMS-APCI calcd. for: C_58_H_39_Br_2_N_6_S_2_ [M+H]^+^ 1043.1018; found: 1043.0968.

*5,15-diphenyl-10,20-bis (2-chloro-10-methyl-10H-phenothiazin-7-yl)-21,23H-porphyrin* 3b 

Purification by column chromatography using dichloromethane/petrol ether (2:1) as eluent gave a purple powder; yield 10% (0.3 g); ^1^H-NMR (400 MHz, CDCl_3_) δ ppm-2.77 (s, 2H, NH), 3.65 (s, 6H), 7.03–7.05 (m, 4H), 7.16–7.21 (m, 4H), 7.76–7.81 (m, 6H), 8.02 (d, 4H_,_
*J* = 7.4Hz), 8.23 (d, 4H, *J* = 8.4Hz), 8.86–8.90 (m, 8H); ^13^C-NMR (100 MHz, CDCl_3_) δ ppm 35.7 (2C), 112.6 (2C), 114.8 (4C), 118.8 (2Cq), 120.2 (2Cq), 121.6 (Cq), 121.7 (2Cq), 122.5 (4C), 126.7 (6C), 127.7 (4Cq), 127.9 (4C), 128.2 (Cq) 132.9 (2Cq), 133.7 (3Cq), 133.9 (3Cq), 134.5 (4C), 136.9 (4C), 142.0 (2Cq), 144.8 (2Cq), 147.0 (4Cq). HRMS-APCI calcd. for: C_58_H_39_Cl_2_N_6_S_2_ [M+H]^+^ 952.1976; found: 952.1905.

#### 3.2.2. General Procedure for the Preparation of Metallo-Phenothiazinyl-Porphyrins

A solution of phenothiazinyl-porphyrin free base (2) or (3) (1 mmol) and Metal(II)-acetate (1.1 mmol) in DMF (50 mL) was heated to reflux for 6 h. The purification was achieved by column chromatography on silica gel, using chloroform/hexane (1:2) as eluent. 


*5,10,15-triphenyl-20-(7-bromo-10-methyl-10H-phenothiazin-3-yl)-21,23-Zn-porphyrin 4a*


Purple powder; yield 90% (0.9 g); ^1^H-NMR (600 MHz, CDCl_3_) δ ppm 3.59 (s, 3H), 6.83 (d, 1H, *J* = 8.7 Hz), 7.15 (d, 1H, *J* = 8.1 Hz), 7.32 (s, 1H), 7.38 (d, 1H, *J* = 8.7 Hz), 7.76–7.82 (m, 9H), 7.97 (s, 1H), 8.02 (d, 1H, *J* = 8.1 Hz), 8.26 (d, 6H, *J* = 8.4 Hz), 8.97–9.01 (m, 6H), 9.01 (d, 2H, *J* = 4.6 Hz); ^13^C-NMR (150 MHz, CDCl_3_) δ ppm 35.7, 112.3, 114.9 (Cq), 115.4, 119.7 (Cq), 120.9 (Cq), 121.2 (2Cq), 125.5 (Cq), 126.5 (6C), 126.6 (2C), 127.5 (3C), 129.6, 130.2, 131.8 (2Cq), 132.0 (6C), 132.1 (2C), 132.8, 133.8, 134.43 (2C), 134.45 (2C), 137.4 (Cq), 142.8 (4Cq), 144.8 (Cq), 145.0 (Cq), 150.2 (6Cq). HRMS-APCI calcd. for: C_51_H_33_BrN_5_ZnS [M+H]^+^ 892.0905; found: 892.0872.


*5,10,15-triphenyl-20-(7-bromo-10-methyl-10H-phenothiazin-3-yl)-21,23-Pd-porphyrin 4b*


Purple powder; yield 80% (0.9 g); ^1^H-NMR (600 MHz, CDCl_3_) δ ppm 3.60 (s, 3H), 6.85 (d, 1H, *J* = 8.4 Hz), 7.15 (d, 1H, *J* = 8.5 Hz), 7.38 (s, 1H, H_6_), 7.40 (d, 1H, *J* = 8.4 Hz), 7.74–7.81 (m, 9H), 7.95–7.98 (m, 2H), 8.17–8.21(m, 6H), 8.81–8.85 (m, 6H), 8.89 (d, 2H, *J* = 4.8 Hz); ^13^C-NMR (150 MHz, CDCl_3_) δ ppm 35.7, 112.3 (Cq), 112.4, 114.3 (Cq), 115.0 (Cq), 115.4, 120.3 (Cq), 121.2 (Cq), 121.7 (2Cq), 125.5 (Cq), 126.71 (4C), 126.78 (2C), 127.4 (Cq), 127.7 (3C), 128.2 (Cq), 129.6, 130.3, 130.8 (2Cq), 130.9 (2C), 131.01 (6C), 131.04 (2C), 132.5, 133.4 (Cq), 133.5, 134.0 (2C), 134.1 (2C), 136.4 (Cq), 141.6 (4Cq), 141.7 (2Cq), 145.0 (Cq). HRMS-APCI calcd. for: C_51_H_33_BrN_5_PdS [M+H]^+^ 934.0648; found: 934.0641.


*5,10,15-triphenyl-20-(7-bromo-10-methyl-10H-phenothiazin-3-yl)-21,23-Ni-porphyrin 4c*


Purple powder; yield 85% (0.9 g); ^1^H-NMR (600 MHz, CDCl_3_) δ ppm 3.57 (s, 3H), 6.82 (d, 1H, *J* = 8.2 Hz), 7.10 (d, 1H, *J* = 7.8 Hz), 7.37 (s, 1H), 7.38 (d, 1H, *J* = 8.2 Hz), 7.69–7.73 (m, 9H), 7.81 (d, 1H, *J* = 7.8 Hz), 7.82 (s, 1H), 8.02–8.05 (m, 6H), 8.76–8.78 (m, 6H), 8.80 (d, 2H, *J* = 4.8 Hz); ^13^C-NMR (150 MHz, CDCl_3_) δ ppm 35.6 (CH_3_), 112.6 (CH), 115.0 (Cq), 115.4 (CH), 117.6 (Cq), 119.0 (2Cq), 121.3 (Cq), 125.5 (Cq), 126.9 (6CH), 127.7 (3CH), 129.6 (CH), 130.3 (CH), 132.0 (2CH), 132.1 (CH), 132.21 (4CH), 132.23 (2CH), 133.28 (CH), 133.7 (6CH), 135.5 (Cq), 140.8 (4Cq), 142.71 (6Cq), 142.74 (2Cq), 145.0 (Cq), 145.1 (Cq) HRMS-APCI calcd. for: C_51_H_33_BrN_5_NiS [M+H]^+^ 886.0967; found: 886.0952.


*5,10,15-triphenyl-20-(7-bromo-10-methyl-10H-phenothiazin-3-yl)-21,23-Cu-porphyrin 4d*


Purple powder; yield 80% (0.86 g), HRMS-APCI calcd. for: C_51_H_33_BrN_5_CuS [M+H]^+^ 889.0930; found: 889.0912.


*5,10,15-triphenyl-20-(8-chloro-10-methyl-10H-phenothiazin-3-yl)-21,23-Zn-porphyrin 5a*


Purple powder; yield 95% (1 g); ^1^H-NMR (600 MHz, CDCl_3_) δ ppm 3.62 (s, 3H), 6.97 (s, 1H), 7.00 (d, 1H, *J* = 8.4 Hz), 7.09 (d, 1H, *J* = 7.9 Hz), 7.16 (d, 1H, *J* = 8.4 Hz), 7.75–7.82 (m, 9H), 7.97 (s, 1H), 8.02 (d, 1H, *J* = 7.9 Hz), 8.23–8.25 (m, 6H), 8.96–8.98 (m, 6H), 9.00 (d, 2H, *J* = 4.6 Hz); ^13^C-NMR (150 MHz, CDCl_3_) δ ppm 35.7, 112.4, 114.7, 119.6 (Cq), 121.21 (2Cq), 121.27 (Cq), 121.3 (Cq), 121.7 (Cq), 122.4, 126.51 (4C), 126.57 (2C), 127.5 (3C), 127.8, 131.8 (3Cq), 132.01 (4C), 132.04 (2C), 132.8, 133.6 (Cq), 133.7, 134.41 (4C), 134.44 (4C), 137.6 (Cq), 142.8 (3Cq), 144.5 (Cq), 147.0 (Cq), 150.2 (5Cq). HRMS-APCI calcd. for: C_51_H_33_ClN_5_ZnS [M+H]^+^ 846.1431; found: 846.1419.


*5,10,15-triphenyl-20-(8-chloro-10-methyl-10H-phenothiazin-3-yl)-21,23-Pd-porphyrin 5b*


Purple powder; yield 90% (1 g); ^1^H-NMR (400 MHz, CDCl_3_) δ ppm 3.64 (s, 3H, CH_3_), 7.02–7.08 (m, 2H, H_1,9_), 7.15 (d, 1H, ^3^*J* = 8.2 Hz, H_3_), 7.34 (d, 1H, *J* = 8.2 Hz H_8_), 7.73–7.79 (m, 9H, H_b,c_), 7.95 (d, 1H, *J* = 8.2 Hz H_4_), 7.98 (s, 1H, H_6_), 8.17–8.20 (m, 6H, H_a_), 8.82–8.84 (m, 6H, H_β_), 8.89 (d, 2H, ^3^*J* = 4.9 Hz, H_β_); ^13^C-NMR (100 MHz, CDCl_3_) δ ppm 35.6, 112.3, 114.3, 120.7 (Cq), 121.7 (2Cq), 121.9 (Cq), 122.8 (Cq), 123.3 (Cq), 126.7 (4C), 126.7 (2C), 127.4, 127.7, 127.7 (3C), 131.0 (4C), 131.0 (2C), 132.5, 133.3, 134.4 (4C), 134.4 (4C), 136.0 (Cq), 141.5 (4Cq), 141.6 (2Cq), 141.7 (2Cq), 141.7 (4Cq), 145.5 (Cq), 145.8 (Cq). HRMS-APCI calcd. for: C_51_H_33_ClN_5_PdS [M + H]^+^ 890.1178; found: 890.1147.


*5,10,15-triphenyl-20-(8-chloro-10-methyl-10H-phenothiazin-3-yl)-21,23-Ni-porphyrin 5c*


Purple powder; yield 88% (0.94 g); ^1^H-NMR (600 MHz, CDCl_3_) δ ppm 3.58 (s, 3H), 6.96–7.02 (m, 2H), 7.11–7.13 (m, 2H), 7.68–7.73 (m, 9H), 7.81–7.83 (m, 2H), 8.03–8.07 (d, 6H), 8.78–8.81 (m, 8H); ^13^C-NMR (150 MHz, CDCl_3_) δ ppm 35.6, 112.7, 114.7, 117.6 (Cq), 119.02 (Cq), 119.07, 121.7 (Cq), 121.8 (Cq), 122.5, 126.8 (6C), 127.7 (3C), 127.9, 132.0 (2C), 132.1, 132.24 (4C), 132.29 (2C), 133.1 (2Cq), 133.7 (6C), 135.7 (Cq), 140.9 (4Cq), 142.7 (8Cq), 144.9 (Cq), 146.9 (Cq). HRMS-APCI calcd. for: C_51_H_33_ClN_5_NiS [M+H]^+^ 840.1493; found: 840.1474.


*5,10,15-triphenyl-20-(8-chloro-10-methyl-10H-phenothiazin-3-yl)-21,23-Cu-porphyrin 5d*


Purple powder; yield 80% (0.86 g); HRMS-APCI Calcd. for: C_51_H_33_ClN_5_CuS [M+H]^+^ 845.1435; found: 845.1420.


*5,15-diphenyl-10,20-bis(7-bromo-10-methyl-10H-phenothiazin-3-yl)-21,23-Zn-porphyrin 6a*


Purple powder; yield 80% (0.85 g); ^1^H-NMR (600 MHz, CDCl_3_) δ ppm 3.52 (s, 6H), 6.78 (d, 2H, *J* = 8.5 Hz), 7.08 (d, 2H, *J* = 8.2 Hz), 7.26 (s, 2H), 7.34 (d, 2H, *J* = 8.5 Hz), 7.75–7.79 (m, 6H), 7.90–7.98 (m, 4H), 8.22–8.24 (m, 4H), 8.94–8.96 (m, 8H); ^13^C-NMR (150 MHz, CDCl_3_) δ ppm 35.6 (2C), 112.2 (2C), 112.3 (Cq), 114.8 (2Cq), 114.9 (Cq), 115.4 (2C), 119.6 (Cq), 120.7 (Cq), 120.8 (Cq), 121.14 (Cq), 121.17 (2Cq), 125.3 (Cq), 125.4 (Cq), 126.53 (2C), 126.57 (2C), 127.4 (2C), 129.5 (2Cq), 129.6 (Cq), 130.2 (2C), 131.8 (2C), 132.03 (2C), 132.08 (2C), 132.8 (Cq), 132.9 (Cq), 133.82 (2Cq), 133.86 (Cq), 134.5 (2C), 137.5 (2C), 142.9 (2C), 144.7 (4Cq), 145.0 (2C), 150.2 (6Cq). HRMS-APCI calcd. for: C_58_H_37_Br_2_N_6_ZnS_2_ [M+H]^+^ 1105.0153; found: 1105.0118.


*5,15-diphenyl-10,20-bis(7-bromo-10-methyl-10H-phenothiazin-3-yl)-21,23-Pd-porphyrin 6b*


Purple powder; yield 80% (0.88 g); ^1^H-NMR (400 MHz, CDCl_3_) δ ppm 3.54 (s, 6H), 6.81 (d, 2H, *J* = 8.8 Hz), 7.01 (d, 2H, *J* = 8.4 Hz), 7.20 (d, 2H, *J* = 8.8 Hz), 7.36 (s, 2H), 7.74–7.78 (m, 6H), 7.90–7.94 (m, 4H), 8.18 (d, 4H, *J* = 8.6 Hz), 8.81–8.87 (m, 8H); ^13^C-NMR (150 MHz, CDCl_3_) δ ppm 35.6 (2C), 112.4 (2C), 114.4 (Cq), 115.0 (2Cq), 115.4 (2C), 120.4 (Cq), 121.9 (Cq), 125.3 (Cq), 125.50 (Cq), 125.54 (Cq), 1263.7 (6C), 127.8 (2C), 129.0 (2C), 129.7 (4C), 130.0 (2C), 130.0 (2C), 130.3 (2C), 131.0 (2C), 131.14 (2C), 131.19 (2C), 132.4 (2Cq), 133.5 (Cq), 134.11 (2Cq), 134.14 (2Cq), 136.3 (Cq), 141.61 (2Cq), 141.67 (2Cq), 141.7 (2Cq), 145.07 (2Cq), 145.09 (2Cq). HRMS-APCI calcd. for: C_58_H_37_Br_2_N_6_PdS_2_ [M+H]^+^ 1146.9871; found: 1146.9896.


*5,15-diphenyl-10,20-bis(7-bromo-10-methyl-10H-phenothiazin-3-yl)-21,23-Ni-porphyrin 6c*


Purple powder; yield 85% (0.89 g); ^1^H-NMR (600 MHz, CDCl_3_) δ ppm 3.55 (s, 6H), 6.80 (d, 2H, *J* = 9.2 Hz), 7.08 (d, 2H, *J* = 8.0 Hz), 7.36–7.37 (m, 4H), 7.68–7.73 (m, 6H), 7.79–7.81 (m, 4H), 8.02 (d, 4H, *J* = 8.4 Hz), 8.76–8.80 (m, 8H); ^13^C-NMR (150 MHz, CDCl_3_) δ ppm 35.6 (2C), 112.6 (2C), 115.0 (2Cq), 115.4 (2C), 117.7 (Cq), 119.1 (Cq), 121.4 (2Cq), 125.4 (2Cq), 126.9 (4C), 127.8 (2Cq), 129.6 (2C), 130.3 (2C), 132.0 (2C), 132.1 (4C), 132.2 (2C), 132.3 (2C), 133.2 (2C), 133.7 (6C), 135.5 (Cq), 140.8 (Cq), 142.70 (5Cq), 142.77 (5Cq), 145.0 (2Cq), 145.1 (2Cq). HRMS-APCI calcd. for: C_58_H_37_Br_2_N_6_NiS_2_ [M+H]^+^ 1099.0215; found: 1099.0195.


*5,15-diphenyl-10,20-bis(7-bromo-10-methyl-10H-phenothiazin-3-yl)-21,23-Cu-porphyrin 6d*


Purple powder; yield 80% (0.84 g); HRMS-APCI calcd. for: C_58_H_37_Br_2_N_6_CuS_2_ [M+H]^+^ 1104.0157; found: 1104.0133.


*5,15-diphenyl-10,20-bis(8-chloro-10-methyl-10H-phenothiazin-3-yl)-21,23-Zn-porphyrin 7a*


Purple powder; yield 95% (1 g); ^1^H-NMR (600 MHz, CDCl_3_) δ ppm 3.59 (s, 6H), 6.95 (s, 2H), 6.99 (d, 2H, *J* = 8.4 Hz), 7.04 (d, 2H, *J* = 8.2 Hz), 7.14 (d, 2H, *J* = 8.4 Hz), 7.76–7.81 (m, 6H), 7.91 (d, 2H, *J* = 8.4 Hz), 7.99 (d, 2H, *J* = 8.2 Hz), 8.22–8.24 (m, 4H), 8.95–8.99 (m, 8H); ^13^C-NMR (150 MHz, CDCl_3_) δ ppm 35.6 (2C), 112.4 (2Cq), 114.7 (4C), 119.74 (Cq), 119.78 (Cq), 121.23 (Cq), 121.28 (Cq), 121.53 (Cq), 121.59 (Cq), 122.4 (4C), 126.5 (4C), 127.5 (2C), 127.8 (2C), 131.8 (2C), 131.9 (2C), 132.0 (4C), 132.7 (Cq), 132.8 (Cq), 133.6 (2Cq), 133.7 (2Cq), 134.4 (2C), 137.5 (2Cq), 142.8 (2Cq), 144.4 (2Cq), 146.9 (2Cq), 150.23 (4Cq), 150.27 (4Cq) HRMS-APCI calcd. for: C_58_H_37_Cl_2_N_6_ZnS_2_ [M+H]^+^ 1017.1154; found: 1017.1118.


*5,15-diphenyl-10,20-bis(8-chloro-10-methyl-10H-phenothiazin-3-yl)-21,23-Pd-porphyrin 7b*


Purple powder; yield 90% (1 g); ^1^H-NMR (600 MHz, CDCl_3_) δ ppm 3.65 (s, 6H), 7.00 (s, 2H), 7.04 (d, 2H, *J* = 8.4 Hz), 7.18–7.21 (m, 4H), 7.75–7.81 (m, 6H), 7.96–7.99 (m, 4H), 8.18 (d, 4H, *J* = 8.6 Hz), 8.82–8.90 (m, 8H); ^13^C-NMR (150 MHz, CDCl_3_) δ ppm 35.6 (2C), 112.6 (2Cq), 114.7 (4C), 119.72, 119.77, 121.2 (2Cq), 121.50 (Cq), 121.53 (Cq), 122.5 (4C), 126.7 (4C), 127.5 (2C), 127.8 (2C), 131.8 (2C), 131.9 (2C), 132.0 (2C), 132.7 (Cq), 132.8 (Cq), 133.6 (2Cq), 133.7 (2Cq), 134.7 (4C), 137.5 (2Cq), 142.8 (2Cq), 144.4 (2Cq), 146.9 (2Cq), 150.2 (4Cq), 150.2 (4Cq). HRMS-APCI calcd. for: C_58_H_37_Cl_2_N_6_PdS_2_ [M+H]^+^ 1059.0931; found: 1059.0891.


*5,15-diphenyl-10,20-bis(8-chloro-10-methyl-10H-phenothiazin-3-yl)-21,23-Ni-porphyrin 7c*


Ni-2j Purple powder; yield 85% (0.9 g); ^1^H-NMR (600 MHz, CDCl_3_) δ ppm 3.53 (s, 6H), 6.93 (s, 2H), 6.98 (d, 2H, *J* = 8.6 Hz), 7.05 (d, 2H, *J* = 8.2 Hz), 7.10 (d, 2H, *J* = 8.6 Hz), 7.68–7.72 (m, 6H), 7.78 (d, 2H, *J* = 8.2 Hz), 7.82 (s, 2H), 8.04 (d, 4H), 8.77–8.81 (m, 8H,); ^13^C-NMR (150 MHz, CDCl_3_) δ ppm 35.6 (2C), 112.6 (2Cq), 114.7 (2C), 119.70, 119.75, 121.2 (2Cq), 121.50 (Cq), 121.53 (Cq), 122.7(4C), 126.8 (4C), 127.3 (2C), 127.9 (2C), 130.7 (2C), 131.4 (2C), 132.5 (2C), 132.9 (2Cq), 133.1 (2Cq), 133.5 (2Cq), 133.6 (2Cq), 134.7 (4C), 137.6 (2Cq), 142.9 (2Cq), 144.4 (2Cq), 146.9 (2Cq), 150.3 (4Cq), 150.4 (4Cq). HRMS-APCI calcd. for: C_58_H_37_Cl_2_N_6_NiS_2_ [M+H]^+^ 1009.1246; found: 1009.1214.


*5,15-diphenyl-10,20-bis(8-chloro-10-methyl-10H-phenothiazin-3-yl)-21,23-Cu-porphyrin 7d*


Purple powder; yield 80% (0.85 g); HRMS-APCI calcd. for: C_58_H_37_Cl_2_N_6_CuS_2_ [M+H]^+^ 1014.1188; found: 1014.1163.

#### 3.2.3. General Procedure for the Synthesis of Suzuki Cross-Coupling Products 

A degassed solution of halogenoMPP porphyrin 2a, 3a or 4a (0.12 mmol *1eq*), aryl-boronic pinacolate ester (0.13 mmol, *1.1eq*), Pd(PPh_3_)_2_Cl_2_ (0.006 mmol *0.05eq*), K_2_CO_3_ (0.1 mmol) in DME (30 mL) and water (15), was heated to reflux under argon atmosphere for 18 h. After cooling to room temperature, the obtained precipitate was filtered off and washed with water. The purification of cross-coupling product was achieved by column chromatography on silica gel using DCM or chloroform as eluent. 


*5,10,15-triphenyl-20-[7-(10′-methyl-10′H-phenothiazine)-10-methyl-10H-phenothiazin-3-yl]-21,23H-porphyrin 8*


By starting with 0.15 g 2a, purification of the reaction product by column chromatography using DCM eluent gave: purple powder; yield 50% (0.07 g); ^1^H-NMR (400 MHz, CDCl_3_) δ ppm -2.76 (s, 2H, NH), 3.64 (s, 6H), 6.87 (d, 1H, *J* = 8.3 Hz), 7.19 (d, 1H, *J* = 7.9 Hz), 7.39–7.42 (m, 3H), 7.69–7.80 (m, 12H), 7.89 (d, 1H, *J* = 8.2 Hz), 8.02 (s, 1H), 8.04 (d, 1H, *J* = 7.9 Hz), 8.12 (d, 1H, *J* = 7.6 Hz), 8.19 (d, 1H, *J* = 7.6 Hz), 8.21–8.25 (m, 6H), 8.86–8.88 (m, 6H), 8.91 (d, 2H, *J* = 4.7 Hz); ^13^C-NMR (100 MHz, CDCl_3_) δ ppm 35.3, 36.8, 112.2 (Cq), 113.7, 114.0, 114.2 (2C), 115.4 (Cq), 118.8 (Cq), 121.15 (Cq), 121.19 (2Cq), 122.5, 122.7, 123.0 (Cq), 123.8 (Cq), 124.9 (2C), 125.4 (2C), 126.4 (4C), 126.5, 127.2 (2C), 127.4 (2C), 127.5 (2C), 129.6 (Cq), 131.8 (Cq), 131.9 (6C), 132.8 (Cq), 134.41 (6C), 134.48 (3C), 137.6 (Cq), 140.5 (Cq), 142.9 (2Cq), 144.7 (4Cq), 145.1 (2Cq), 145.6 (2Cq), 148.6 (4Cq). HRMS-APCI calcd. for: C_64_H_45_N_6_S_2_ [M+H]^+^ 961.3068; found: 961.1229.


*5,10,15-triphenyl-20-(7-(10′-methyl-10′H-phenothiazine)-10-methyl-10H-phenothiazin-3-yl)-21,23-Zn-porphyrin 8a*


By starting with 0.15 g 4a, purification of the reaction product by column chromatography using DCM eluent gave: purple powder; yield 55% (0.09 g); ^1^H-NMR (400 MHz, CDCl_3_) δ ppm 3.39 (s, 3H), 3.60 (s, 3H), 6.81–6.86 (m, 3H), 6.93 (t, 1H, *J* = 7.4 Hz), 7.13 (s, 1H), 7.17–7.21 (m, 2H), 7.29 (d, 1H, *J* = 7.4 Hz), 7.31 (s, 1H), 7.34 (d, 1H, *J* = 7.6 Hz), 7.38 (d, 1H, *J* = 7.4 Hz), 7.76–7.80 (m, 9H), 7.98 (s, 1H), 8.01 (d, 1H, *J* = 7.8 Hz), 8.25 (d, 6H), 8.99 (d, 6H), 9.00 (d, 1H, *J* = 4.5 Hz), 9.03 (d, 1H, *J* = 4.5 Hz); ^13^C-NMR (100 MHz, CDCl_3_) δ ppm 35.3, 40.2, 112.2 (Cq), 114.0 (2C), 114.2 (2C), 115.4 (Cq), 118.8 (Cq), 121.11 (Cq), 121.14 (2Cq), 122.5 (2C), 123.0 (Cq), 123.8 (Cq), 124.9 (2C), 125.4 (2C), 126.4 (4C), 126.5, 127.2 (2C), 127.4 (2C), 127.5 (2C), 129.6 (Cq), 131.8 (Cq), 131.9 (6C), 132.8 (Cq), 134.4 (4C), 134.4 (3C), 137.6 (Cq), 142.9 (2Cq), 144.7 (Cq), 145.1 (Cq), 145.6 (2Cq), 150.2 (8Cq). HRMS-APCI calcd. for: C_64_H_42_N_6_S_2_Zn [M+H]^+^ 1022.2203; found: 1022.1968.


*5,10,15-triphenyl-20-(7-(3′-formyl-10′-methyl-10H-phenothiazine)-10-methyl-10H-phenothiazin-3-yl)-21,23H-porphyrin 9*


When starting with 0.1 g 2a, purification of the reaction product by column chromatography with DCM gave a purple powder; yield 60% (0.07 g); ^1^H-NMR (600 MHz, CDCl_3_) δ ppm -2.76 (s, 2H, NH), 3.49 (s, 3H), 3.60 (s, 3H), 6.88–6.91 (m, 4H), 7.06–7.10 (m, 4H), 7.63–7.66 (m, 2H), 7.76–7.79 (m, 9H), 8.02–8.05 (m, 2H), 8.24 (d, 6H, *J* = 7.8 Hz), 8.87–8.93 (m, 8H), 9.83 (s, 1H, CHO); ^13^C-NMR (150 MHz, CDCl_3_) δ ppm 35.6, 35.8, 112.4, 113.6 (2C), 114.5, 115.0 (2C), 119.0 (2Cq), 120.2, 121.50 (Cq), 121.53 (Cq), 123.0, 123.6 (2Cq), 123.8 (2Cq), 125.0 (3C), 125.2 (2C), 125.7 (6C), 126.7 (4C), 127.7, 127.9, 130.5 (2C), 131.1 (2C), 133.0, (2Cq) 133.9 (2Cq), 134.2, 134.5 (4C), 135.5 (2Cq), 136.6, 142.1 (4Cq), 143.0 (2Cq), 145.0 (2Cq), 145.2 (3Cq), 150.7 (2Cq), 190.0 (CHO); HRMS (APCI) calcd. for: C_65_H_45_N_6_OS_2_ [M+H^+^] 989.3018; found: 989.3071. 


*5,10,15-triphenyl-20-[7-(4-(9H-carbazole-9-yl)phenyl)-10-methyl-10H-phenothiazin-3-yl]-21,23H-porphyrin 10*


When starting with 0.15 g 2a, purification of the reaction product by column chromatography with chloroform/petrol ether 1:1 eluent gave: purple powder; yield 70% (0.12 g); ^1^H-NMR (400 MHz, CDCl_3_) δ ppm -2.75 (s, 2H, NH), 3.72 (s, 3H), 7.13 (d, 1H, *J* = 8.7 Hz), 7.21–7.23 (m, 1H), 7.31–7.35 (m, 3H), 7.44–7.47 (m, 3H, H), 7.51 (d, 2H, *J* = 8.1 Hz), 7.61 (m, 2H), 7.65 (d, 2H, *J* = 8.1 Hz), 7.72–7.80 (m, 9H), 8.03 (d, 1H, *J* = 8.0 Hz), 8.07 (s, 1H), 8.18 (d, 2H, *J* = 7.7 Hz), 8.24 (d, 6H, *J* = 7.5 Hz), 8.87 (s, 6H), 8.95 (d, 2H, ^3^*J* = 4.4 Hz); ^13^C-NMR (100 MHz, CDCl_3_) δ ppm 35.8, 109.8 (2C), 112.4 (2Cq), 114.7 (2Cq), 118.9 (2Cq), 119.9 (2C), 120.2, 121.5, 123.4 (2C), 125.9 (6C), 126.7 (9C), 127.4 (4C), 127.7 (2C), 127.9 (4C), 133.0 (2Cq), 134.0 (2Cq), 134.5 (8C), 135.0 (2Cq), 136.6 (2Cq), 136.7 (2Cq), 140.8 (4Cq), 142.1 (4Cq), 145.2 (Cq), 145.4 (2Cq); HRMS (APCI) calcd. for: C_69_H_47_N_6_S [M+H^+^] 991.3577; found: 991.3573. 


*5,10,15-triphenyl-20-(7-phenyl-10-methyl-10H-phenothiazin-3-yl)-21,23H-porphyrin 11*


By starting with 0.1 g 2a, purification of the reaction product by column chromatography using DCM eluent gave: purple powder; yield 60% (0.06 g); ^1^H-NMR (400 MHz, CDCl_3_) δ ppm -2.77 (s, 2H, NH), 3.65 (s, 3H), 7.04 (d, 1H, *J* = 8.5 Hz), 7.16 (d, 1H, *J* = 8.1 Hz), 7.30–7.33 (m, 1H, H), 7.42 (t, 2H, *J* = 7.4 Hz, *J* = 7.7 Hz), 7.50 (d, 2H, *J* = 7.6 Hz), 7.57 (d, 2H, *J* = 7.4 Hz), 7.72–7.75 (m, 9H), 7.99 (d, 1H, *J* = 8.1 Hz), 8.02 (s, 1H), 8.20 (d, 6H, *J* = 6.3 Hz), 8.83 (s, 6H), 8.90 (d, 2H, ^3^*J* = 4.5 Hz); ^13^C-NMR (100 MHz, CDCl_3_) δ ppm 35.9, 112.5, 114.8, 118.8 (Cq), 120.2 (2Cq), 121.4 (Cq), 124.1 (Cq), 126.0, 126.7 (6C), 126.9 (4C), 127.6, 127.7 (3C), 129.6 (Cq), 129.7 (7C), 130.3 (Cq), 130.3 (4C), 131.0 (Cq), 133.0, 133.7 (Cq), 134.0, 134.1 (Cq), 134.2 (Cq), 134.5 (4C), 134.9 (Cq), 136.9 (Cq), 142.1 (4Cq), 144.9 (Cq), 145.8 (Cq), 146.1 (Cq), 150.2 (Cq), 165.4 (Cq); HRMS (APCI) calcd. for: C_57_H_40_N_5_S [M+H^+^] 826.8334; found: 826.8308. 


*5,10,15-triphenyl-20-(7-phenyl-10-methyl-10H-phenothiazin-3-yl)-21,23-Zn-porphyrin 11a*


By starting with 0.15g 4a, purification of the reaction product by column chromatography using DCM eluent gave a purple powder; yield 66% (0.1 g); ^1^H-NMR (600 MHz, CDCl_3_) δ ppm 3.37 (s, 3H), 6.87 (d, 1H, *J* = 8.7 Hz), 7.10 (d, 1H, *J* = 8.5 Hz), 7.21 (d, 1H, *J* = 8.1 Hz), 7.36–7.37 (m, 1H), 7.41 (d, 1H, *J* = 8.6 Hz), 7.46 (t, 1H, *J* = 7.3 Hz, *J* = 8.0 Hz), 7.51 (s, 1H), 7.55 (d, 1H, *J* = 6.4 Hz), 7.61 (d, 1H, *J* = 7.4 Hz), 7.76–7.80 (m, 9H), 8.03–8.04 (m, 2H), 8.24–8.25 (m, 6H), 8.97 (s, 6H, H_β_), 9.01 (d, 1H, *J* = 4.5 Hz), 9.04 (d, 1H, *J* = 4.5 Hz); ^13^C-NMR (150 MHz, CDCl_3_) δ ppm 37.6, 112.9, 115.8, 119.8 (Cq), 120.8 (2Cq), 121.4 (Cq), 123.7, 124.1 (Cq), 126.4 (6C), 126.6 (4C), 128.2, 128.7 (3C), 129.6 (Cq), 129.7 (7C), 130.5 (Cq), 131.9 (4C), 132.0, 133.7 (Cq), 133.8 (Cq), 134.0, 134.1 (Cq), 134.3 (Cq), 134.4 (4C), 137.3 (Cq), 139.2 (Cq), 142.8 (Cq), 145.5 (Cq), 146.4 (Cq), 148.8 (Cq), 150.2 (Cq), 150.4 (2Cq), 150.7 (Cq), 156.6 (Cq); HRMS (APCI) calcd. for: C_57_H_38_N_5_SZn [M+H^+^] 888.2134; found: 888.2108. 


*5,10,15-triphenyl-20-(7-(4-formyl-phenyl)-10-methyl-10H-phenothiazin-3-yl)-21,23H-porphyrin 12*


By starting with 0.15 g 2a, purification of the reaction product by column chromatography using DCM eluent gave: purple powder; yield 40% (0.04 g); ^1^H-NMR (600 MHz, CDCl_3_) δ ppm -2.74 (s, 2H, NH), 3.71 (s, 3H), 7.09 (d, 1H, *J* = 8.4 Hz), 7.22 (d, 1H, *J* = 7.8 Hz), 7.57–7.59 (m, 2H), 7.78–7.81 (m, 9H), 7.95 (d, 2H, *J* = 7.9 Hz), 8.04–8.06 (m, 2H), 8.19 (s, 1H), 8.24 (d, 6H, *J* = 6.4 Hz), 8.83 (d, 1H, *J* = 6.3 Hz), 8.87–8.93 (m, 8H), 10.06 (s, 1H, CHO); ^13^C-NMR (150 MHz, CDCl_3_) δ ppm 35.8, 112.5, 114.6, 118.8 (Cq), 120.2 (2Cq), 121.4 (Cq), 124.1 (Cq), 126.0, 126.7 (6C), 126.9 (4C), 127.7 (3C), 129.6 (Cq), 130.3 (Cq), 129.7 (7C), 130.3 (4C), 131.0 (Cq), 133.0, 133.7 (Cq), 134.0, 134.1 (Cq), 134.2 (Cq), 134.5 (4C), 134.9 (Cq), 136.9 (Cq), 142.1 (4Cq), 144.9 (Cq), 145.8 (Cq), 146.1 (Cq), 150.2 (Cq), 165.4 (2Cq), 191.8 (CHO); HRMS (APCI) calcd. for: C_58_H_40_N_5_OS [M+H^+^] 854.2948; found: 854.2949. 


*5,15-diphenyl-15,20-bis(7-(3′-formyl-10′-methyl-10′H-phenothiazin-7yl)-10-methyl-10H-phenothiazin-3-yl)-21,23H-porphyrin 13*


After starting with 0.1g 3a, purification of the reaction product by column chromatography with DCM gave 13 as a purple powder; yield 65% (0.1 g); ^1^H-NMR (400 MHz, CDCl_3_) δ ppm −2.75 (s, 2H, NH), 3.43 (s, 6H), 3.65 (s, 6H), 6.85 (s, 4H), 7.03 (d. 2H, *J* = 6.2 Hz), 7.18 (d, 2H, *J* = 8.0 Hz), 7.37–7.39 (m, 4H), 7.45 (s, 4H), 7.63–7.66 (m, 4H), 7.77–7.79 (m, 6H), 8.01 (d, 2H, ^3^*J* = 8.0 Hz), 8.04 (s, 2H), 8.23 (d, 4H, *J* = 6.2 Hz), 8.87–8.92 (m, 8H), 9.81 (s, 2H, CHO); ^13^C-NMR (100 MHz, CDCl_3_) δ ppm 35.7 (2C), 35.8 (2C), 112.4 (2Cq), 113.7 (4C), 114.6 (2Cq), 115.0 (4C), 119.0 (2Cq), 120.2 (2Cq), 121.5 (Cq), 123.0 (4C), 123.6 (2Cq), 123.9 (Cq), 125.1 (4C), 125.3 (2Cq), 125.7 (6C), 126.7 (4C), 127.7 (2C), 128.0 (2C), 130.5 (2Cq), 131.1 (4C), 133.0 (Cq), 134.0 (Cq), 134.3 (2Cq), 134.6 (4C), 135.5 (2C), 136.6 (2C), 142.1 (4Cq), 143.0 (4Cq), 145.0 (4Cq), 145.2 (2Cq), 150.8 (4Cq), 190.1 (2CHO); HRMS (APCI) calcd. for: C_86_H_59_N_8_O_2_S_4_ [M+H^+^] 1363.3638; found: 1363.3625. 


*5,15-diphenyl-15,20-bis [7-(4-(9H-carbazole-9-yl)phenyl)-10-methyl-10H-phenothiazin-3-yl)]-21,23H-porphyrin 14*


By starting with 0.1 g 3a, purification of the reaction product by column chromatography with chloroform/petrol ether 1:1 eluent gave a purple powder; yield 70% (0.1 g); ^1^H-NMR (600 MHz, CDCl_3_) δ ppm -2.72 (s, 2H, NH), 3.72 (s, 6H), 7.12 (d, 2H, *J* = 9.0 Hz), 7.22 (d, 2H, *J* = 8.1 Hz), 7.33 (t, 4H, *J* = 7.4 Hz, *J* = 7.2 Hz), 7.46 (t, 4H, *J* = 7.9 Hz, *J* = 7.2 Hz), 7.50–7.52 (m, 4H), 7.61–7.62 (m, 2H), 7.62 (s, 2H), 7.66 (d, 4H, *J* = 8.3 Hz), 7.78–7.83 (m, 10H), 8.05 (s, 2H), 8.09 (s, 2H), 8.10 (d, 4H, *J* = 7.6 Hz), 8.25 (s, 4H), 8.88–8.96 (m, 8H); ^13^C-NMR (150 MHz, CDCl_3_) δ ppm 35.7 (2C), 109.8 (6C), 112.4 (2Cq), 114.7 (2C), 119.0 (Cq), 119.9 (2Cq), 120.0 (4C), 120.2 (2Cq), 120.3 (4C), 121.5 (Cq), 123.4 (6C), 124.0 (2Cq), 125.9 (2C), 125.9 (6C), 126.4 (2Cq), 126.7 (2C), 126.7 (2C), 127.4 (8C), 127.7 (2C), 127.7 (2C), 127.9 (8C), 133.0 (Cq), 133.0 (Cq), 134.0 (Cq), 134.0 (Cq), 134.6 (2Cq), 135.0 (Cq), 136.6 (2Cq), 136.7 (Cq), 139.0 (2Cq), 140.8 (8Cq), 142.1 (2Cq), 145.2 (2Cq), 145.4 (2Cq); HRMS (APCI) calcd. for: C_94_H_63_N_8_S_2_ [M+H^+^] 1368.4645; found: 1368.4595.


*5,15-diphenyl-15,20-bis(7-(4-formyl-phenyl)-10-methyl-10H-phenothiazin-3-yl)-21,23H-porphyrin 15*


Purification by column chromatography with DCM gave: purple powder; yield 50% (0.06 g); ^1^H-NMR (600 MHz, CDCl_3_) δ ppm -2.76 (s, 2H, NH), 3.72 (s, 6H), 7.11 (d, 2H, *J* = 8.0 Hz), 7.23 (d, 2H, *J* = 7.1 Hz), 7.57 (s, 2H), 7.60 (d, 2H, *J* = 8.5 Hz), 7.77–7.81 (m, 10H), 7.96 (d, 4H, *J* = 8.0 Hz), 8.04–8.05 (m, 4H), 8.22–8.24 (m, 4H), 8.86–8.93 (m, 8H), 10.06 (s, 2H, CHO); ^13^C-NMR (150 MHz, CDCl_3_) δ ppm 35.8 (2C), 112.5 (2Cq), 112.6 (2Cq), 114.6 (2Cq), 118.8 (Cq), 120.3 (2Cq), 124.0 (Cq), 126.1 (2C), 126.7 (2C), 126.7 (2C), 126.9 (6C), 127.0 (2C), 127.2 (2Cq), 127.7 (2C), 130.4 (4C), 130.4 (2C), 133.0 (2C), 134.0 (2C), 134.2 (2C), 134.5(4C), 134.9 (4C), 136.8 (2C), 142.0 (2Cq), 144.9 (4Cq), 145.8 (4Cq), 146.2 (4Cq), 175.6 (4Cq), 191.8 (2CHO); HRMS (APCI) calcd. for: C_72_H_49_N_6_O_2_S_2_ [M+H^+^] 1093.3352; found: 1093.3356.


*5,10,15,20-tetrakis-[4-(3-formyl-10-methyl-10Hphenothiazin-7-yl)phenyl]-21,23H-porphyrin 17*


A mixture of 0.1 g BrTPP, 0.23 g 3-formyl-10-methyl-10Hphenothiazin-7-yl boronic pinacolate, 0.025 g Pd(PPh_3_)_2_Cl_2_, 0.12 g K_2_CO_3_, 30 mL DME and 15 mL H_2_O was refluxed for 18 h. Purification by column chromatography with DCM gave a purple powder; yield 64% (0.1 g); ^1^H-NMR (600 MHz, CDCl_3_) δ ppm –2.72 (s, 2H, NH), 3.58 (s, 12H), 6.97 (d, 4H, *J* = 8.2 Hz), 7.08 (d, 4H, *J* = 8.6 Hz), 7.18–7.20 (m, 4H), 7.72–7.77 (m, 12H), 7.95 (d, 8H, *J* = 7.9 Hz), 8.11–8.13 (m, 4H), 8.28–8.31 (m, 4H), 8.85–8.95 (m, 8H), 9.87 (s, 4H, CHO); HRMS (APCI) calcd. for: C_100_H_67_N_8_O_4_S_4_ [M+H^+^] 1572.4196; found: 1572.3790.


*5,10,15,20-tetrakis-(4′-(9H-carbazole-9yl)-[1,1′-biphenyl]-4yl)-21,23H-porphyrin 18*


A mixture of 0.1 g BrTPP, 0.17 g (4-(9*H*-carbazol-9-yl)phenyl)boronic acid, 0.025 g Pd(PPh_3_)_2_Cl_2_, 0.12 g K_2_CO_3_, 30 mL DME and 15 mL H_2_O was refluxed for 18 h. Purification by column chromatography with DCM gave a purple powder; yield 65% (0.1 g); ^1^H-NMR (600 MHz, CDCl_3_) δ ppm -2.70 (s, 2H, NH), 7.38 (t, 6H, *J* = 7.2 Hz, *J* = 7.4 Hz), 7.53 (t, 6H, *J* = 7.5 Hz, *J* = 7.5 Hz), 7.62 (d, 4H, *J* = 8.1 Hz), 7.83 (d, 8H, *J* = 7.8 Hz), 7.91–7.94 (m, 6H), 8.09–8.12 (m, 10H, H), 8.14 (d, 8H, *J* = 7.9 Hz), 8.24 (d, 8H, *J* = 7.8 Hz), 8.38 (d, 4H, *J* = 7.5 Hz), 8.87–8.91 (m, 8H), 9.03 (s, 4H); ^13^C-NMR (150 MHz, CDCl_3_) δ ppm 109.9 (8C), 120.1 (16Cq), 120.4 (8C), 123.5 (8C), 125.4 (4Cq), 126.0 (16C), 127.6 (8C), 128.7 (16C), 130.0 (8C), 135.2 (4Cq), 135.8 (4Cq), 135.8 (4Cq), 140.9 (12Cq); HRMS (APCI) calcd. for: C_116_H_75_N_8_ [M+H^+^] 1580.6142; found: 1580.6138.


*5,10,15,20-tetrakis-(4′-formyl-[1,1′-biphenyl]-4yl)-21,23H-porphyrin **19***


A mixture of 0.1 g BrTPP, 0.015 g 4-formylphenylboronic acid, 0.025 g Pd(PPh_3_)_2_Cl_2_, 0.12 g K_2_CO_3_, 30 mL DME and 15 mL H_2_O was refluxed for 18 h. Purification by column chromatography with DCM gave a purple powder; yield 48% (0.05 g); ^1^H-NMR (600 MHz, CDCl_3_) δ ppm 2.65 (s, 2H, NH), 8.09 (d, 8H, *J* = 7.6 Hz), 8.12–8.16 (m, 16H), 8.38 (d, 8H, *J* = 7.7 Hz), 8.98 (s, 8H), 10.19 (s, 4H, CHO); HRMS (APCI) calcd. for: C_72_H_47_N_4_O_4_ [M+H^+^] 1031.3591; found: 1031.3591.

### 3.3. In Vitro Biological Assay

The synthesized phenothiazine-bridged porphyrin-(hetero)aryl dyads with fluorescence properties selected for the biological assay which were simply labeled in this section as dyes were as follows:


*5,10,15-triphenyl-20-(7-(3′-formyl-10′-methyl-10H-phenothiazine)-10-methyl-10H-phenothiazin-3-yl)-21,23H-porphyrin 9*


*5,15-diphenyl-15,20-bis(7-(3′-formyl-10′-methyl-10′H-phenothiazin-7yl)-10-methyl-10H-phenothiazin-3-yl)-21,23H-porphyrin* 13


*5,15-diphenyl-15,20-bis(7-(4-formyl-phenyl)-10-methyl-10H-phenothiazin-3-yl)-21,23H-porphyrin **15***


The general equipment for the in vitro testing consisted of: a class II biological safety cabinet (from ESCO Micro Pte. Ltd., Singapore), an incubator with 37 °C and 5% CO_2_ parameters (Revco Ultima II, from Thermo Electron Corporation, Asheville, NC, USA), a centrifuge with swing-out rotors (32R from Hettich Lab Technology, Tuttlingen, Germany), an inverted phase microscope CKX41 (Olympus Life Sciences, Rockville, MD, USA) and a multiplate reader (Synergy2 from BioTek Company, Winooski, VT, USA).

The human ovarian tumor cell lines A2780 and A2780cis were acquired from the European Collection of Authenticated Cell Cultures (ECACC, Salisbury, UK) and the OVCAR-3 cell line was acquired from the American Type Culture Collection (ATCC, through LGC Standards GmbH, Wesel, Germany). All cell lines were grown in RPMI-1640 cell culture media. For A2780 and A2780cis cell lines the media was supplemented with 10% fetal bovine serum (FBS), while for OVCAR-3 cells 20% FBS was employed together with 0.01 mg/mL insulin from porcine pancreas (all media and supplements were from Sigma-Aldrich Chemie GmbH, Taufkirchen, Germany).

For the cytotoxicity testing, the human ovarian tumor cell lines were harvested at subconfluency and plated on 96-well microplates at a concentration of 2.5 × 10^4^ cells/well in 190 μL cell culture media, and after 24 h they were treated with the tested dyes 9, 13 and 15. Each dye was dissolved in dimethyl sulfoxide (provided by Merck, Darmstadt, Germany) and then eight serial solutions were prepared in phosphate buffered saline solution (PBS, from Sigma Aldrich). In each well, 10 µL of solution was distributed (8 serial concentrations for each dye) and the treated cells were incubated for 24 h. The final concentrations of the compounds in the cell media were in the range of 1.95–250.00 µM for 15, 1.88–240 µM for 9 and 1.64–210 µM for 13. On every plate, wells untreated with dyes were kept as references, and wells containing cell culture media with 9, 13 and 15 solutions (without cells) were kept as color controls for each concentration. Three independent tests were performed for each dye.

The microplates containing the treated human ovarian tumor cells were subjected to cytotoxicity testing using the 3-(4,5-dimethylthiazol-2-yl)-2,5-diphenyltetrazolium bromide viability dye (MTT, from Sigma Aldrich), following a colorimetric protocol described before [29]. The optical densities of the sample supernatants, reflecting the cells’ viability, were recorded, and the survival rate was calculated relative to the untreated control cell populations.

The metabolic rate of the treated human ovarian tumor cells was evaluated by the Alamar Blue test (from Life Technologies Corporation, Eugene, OR USA), as described before [30]. Briefly: the cells were seeded on 96-well plates at the same density as for MTT testing, and incubated for 24 h with 8 different concentrations (from 1.95 to 250 µM) of tested dyes 9, 13 and 15, respectively, in triplicates. The data provided by the fluorescence assessment were used to build a linear regression of dose–response relationship for each tested compound, and to quantify the results with the mathematic parameter hillslope (GraphPad Prism Software La Jolla, CA, USA).

For the cellular uptake analysis by fluorescence microscopy, the human ovarian carcinoma cells were plated on 50mm µ-dish cell culture vessels (from Ibidi GmbH, Gräfelfing, Germany), at a concentration of 10^5^ cells/mL in cell culture media, and the dishes were treated with 50 µM 9, 13 and 15 respectively and further kept for 24 h. Untreated cell culture vessels were kept as references. After the treatment, the cell culture media were removed, and the vessels were washed 3 times with PBS and fixed with a 4% paraformaldehyde (from Sigma Aldrich Chemie, Taufkirchen, Germany) solution in PBS. The µ-dishes were stained with 0.002 µg/mL 4′,6-diamidino-2-phenylindole solution in PBS (DAPI, from Serva Electrophoresis GmbH, Heidelberg, Germany), washed 2 times, and kept covered with PBS before the fluorescence microscope evaluation.

### 3.4. Fluorescence Imaging of Human Ovarian Tumor Cell Lines

The fluorescence images of A2780cis cells treated with 13 were collected with an inverted Axio Observer Z1 microscope (from Carl Zeiss), using a Compact Light Source HXP 120 C mercury lamp. Briefly, the light was reflected by a dichroic mirror using G 365 excitation filter (i.e., filter set 49 from Carl Zeiss) in order to visualize the cell nuclei, and an excitation filter BP 470/40 (i.e., filter set 38 from Carl Zeiss) was employed to image the localization of dye 13. The bright-field and fluorescence images were captured using a 63x iris oil immersion objective and a Zeiss AxioCam MRm monochrome microscope camera and then processed using the ZEN software.

Fluorescence lifetime imaging microscopy under two-photons excitation (TPE-FLIM) images on the A2780 cells post 24 h treatment were performed using a MicroTime 200 time-resolved confocal fluorescence microscope system (IX 71, Olympus) and coupled to a Mira 900 Titanium:Sapphire tunable femtosecond laser (from Coherent). The incubated A2780 cells were imaged under the microscope at 800 nm laser beam using up to 25 mW laser power, using the same system configuration described previously [42]. The signal was subsequently collected using a Plan N 40x objective with numerical aperture (NA) of 0.65 and spectrally filtered by a FF01-750 SP emission filter (Semrock, USA). The TPE-FLIM images were acquired and analyzed by the SymPhoTime software provided by PicoQuant. Bright-field images of the A2780 cells scanned through TPE-FLIM were collected on the same microscope, using an Olympus IX-2LW UCD condenser and an Olympus CAM-XC30 digital camera.

## 4. Conclusions

The family of *meso*-phenothiazinyl-phenyl-porphyrin MPP dyes was substantially extended by synthesizing the new halogen-MPP of AB_3_-type (containing one peripheral halogenophenothiazine unit) and the *trans*-A_2_B_2_ type (two peripheral halogenophenothiazine units) further employed, on the one hand, as ligands in new metal-complexes, and on the other hand, as scaffolds for peripheral modification by Suzuki–Miyaura cross coupling with various (hetero)arylboronic acid derivatives. The cross-coupling methodology applied afforded a series of *meso*-phenothiazine-bridged porphyrin-(hetero)aryl dyads in good yields (40–70%). The examination of the UV–Vis optical properties of the newly synthesized compounds emphasized the absorption/emission properties of the porphyrin chromophore slightly modulated by the substitution pattern. In comparison with *para*-phenylene, substitution of phenothiazine in positions 3,7 gave a less effective bridge for the extension of the π-electron system of the novel porphyrine-(hetero)aryl dyads. Increasing the size of the peripheral substituents did not improved the absorption characteristics of MPP dyes in the visible range, but it emphasized attractive NIR fluorophore properties for the symmetrical *trans* A_2_B_2_ type phenothiazine-bridged porphyrin-phenothiazine dyad 13.

According to cell viability colorimetric assays (MTT and Alamar Blue), the phenothiazine-bridged porphyrin-phenothiazine dyads 9, 13 and 15 exhibit a moderate cytotoxicity on ovarian tumor cell lines in vitro.

The green emissive dye 13 can be successfully employed for living human ovarian tumor cell staining, at 50 µM presenting a good cell internalization without affecting the cell viability and having the capability to stain the cells’ nuclei. Visualization of the stained cells can be performed both by fluorescence imaging and TPE-FLIM imaging.

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
