# Peer review of "Novel Phenothiazine-Bridged Porphyrin-(Hetero)aryl dyads: Synthesis, Optical Properties, In Vitro Cytotoxicity and Staining of Human Ovarian Tumor Cell Lines"

_ijms, 2020, doi:10.3390/ijms21093178_

Round 1

Reviewer 1 Report

Castelia Cristea and coworkers report synthesis and optical properties of phenothiazine bridged porphyrin-(hetero)aryl dyads.
And, in vitro cytotoxicity and staining properties of these compounds are also investigated.
The synthesis (2.1) and optical properties (2.2) parts are well written, although the synthetic methods are already known.
The authors demonstrate the vitro cytotoxicity and staining properties of compound 9 (A3B type), 13, and 15 (transA2B2-type).
But, the result with compound 17 (A4-type), or compound 19 (porphyrin without phenothiazine unit) or TPP (commercially available compound) is not shown.
Comparisons are important in such experiments. It is not possible to assess the effects of introduction of phenothiazine unit in peripheral region with the presented data.
In my opinion, this work would be half-baked. Therefore, I think it is not suitable for publication in International Journal of Molecular Sciences.

Author Response

The authors are thankful to the reviewers for the detailed analysis signalizing the weaker parts of our manuscript. We found the reviewers comments very helpful for improving the quality of the manuscript.

Comments and Suggestions for Authors:

Castelia Cristea and coworkers report synthesis and optical properties of phenothiazine bridged porphyrin-(hetero)aryl dyads. 
And, in vitro cytotoxicity and staining properties of these compounds are also investigated.
The synthesis (2.1) and optical properties (2.2) parts are well written, although the synthetic methods are already known.
The authors demonstrate the vitro cytotoxicity and staining properties of compound 9 (A3B type), 13, and 15 (transA2B2-type).
Comments: But, the result with compound 17 (A4-type), or compound 19 (porphyrin without phenothiazine unit) or TPP (commercially available compound) is not shown. 
Answer: The paper describes the use of the newly synthesized phenothiazine bridged porphyrin-(hetero)aryl dyads 9, 13 and 15 as fluorescent dyes for the staining of living ovarian cells and their visualization by fluorescence microscopy techniques. Dyes 17 and 19 did not display fluorescence emissions and for this reason were not included in the biological evaluation study as suggested by the reviewer.

Comments: Comparisons are important in such experiments. It is not possible to assess the effects of introduction of phenothiazine unit in peripheral region with the presented data.
In my opinion, this work would be half-baked. Therefore, I think it is not suitable for publication in International Journal of Molecular Sciences.

Answer: as suggested by the reviewer, a comparison between the fluorescence properties of the synthesized compounds was introduced in the revised manuscript in the text lines 199-214 as follows:

“The presence of the phenothiazine units in dyes scaffolds, may be considered as responsible on one hand for the energy dissipation by typical butterfly vibrations thus inducing low fluorescence emission in solution, but on the other hand, for enlarging the intermolecular distance due to its bent structure thus preventing the intermolecular π-π stacking and thereby stimulating fluorescence in aggregated state [27].

In solution, the parent MPP dyes displayed fluorescence emissions typical to the porphyrin core in 660-670 nm range [4], while the extension of the pending phenothiazine arms of A3B type and A2B2 type MPP with additional peripheral phenothiazine or phenyl units produced fluorescence quenching in compounds 8, 11. The presence of formyl auxochromic group in peripheral positions appeared beneficial by imparting molecular polarization by electron withdrawing effect and thus fluorescence emission was recorded for compounds 9, 12, 13, 15 as shown in table 3. Phenothiazine bridged porphyrin-carbazole dyads A3B type 10 and A2B2 type 14 displayed fluorescence emissions in similar range. The symmetrical A4 type phenylene bridged porphyrin-(hetero)aryl dyads did not display fluorescence emission regardless the nature of the peripheral units: formyl-phenothiazine 17, N-phenyl carbazole 18, or phenyl-carbaldehyde 19.”

And regarding the fluorophores in relation to their structure was introduced the text in lines 273-279 as follows:

“The typical chemical reactivity of the carbaldehyde unit was the selected structural feature meant for passing on the affinity towards the bio-materials. In phenothiazine bridged porphyrin-phenothiazine dyads A3B type 9 and A2B2 type 13 the formyl group was attached to the bent phenothiazine unit(s) prone to prevent the intermolecular π-π stacking and green fluorescence was displayed by these staining agents in aggregated state. Dye 15 containing the formyl group attached to peripheral flat phenylene units exhibited aggregation-caused fluorescence quenching which limited its application”.

Reviewer 2 Report

The authors report synthesis and spectroscopic characterization of a vast amount of porphyrin compounds, yet only three of them were selected for biological assessments. What was the idea behind the synthesis of Zn-, Cu-, Ni- and Pd-complexes of the porphyrins?

Two other remarks:

1) Porphyrins 6 and 7 differ only in a halogen atom situated well afar from the porphyrin chromophore; yet, the absorption spectra of 6c/7c and 6d/7d pair demonstrate different patterns: one Q-band for 7c and 7d vs. two Q-bands for 6c and 6d. That looks strange. Could the authors explain that?

2) Fig. 2 does not look consistent with Table 3. According to table 3, the emission quantum yield for 9 is twice that for 14, while the emission quantum yield for 13 is only 20% larger that of 9. However, the emission intensity shown on Fig. 2 gives another impression of the relative quantum yields of 9, 13 and 14: the emission of 13 is at least two times stronger than that of 9. Are the emission spectra on Fig. 2 recorded for the same absorbance at the excitation wavelength? 

Author Response

The authors are thankful to the reviewers for the detailed analysis signalizing the weaker parts of our manuscript. We found the reviewers comments very helpful for improving the quality of the manuscript.

Comments and Suggestions for Authors

Comments: The authors report synthesis and spectroscopic characterization of a vast amount of porphyrin compounds, yet only three of them were selected for biological assessments.

Answer: Aiming the distinct visualization of living cells using fluorescence microscopy techniques, the selection of candidates for the biological assessments was narrowed by the examination of optical properties of the newly synthesized compounds. The selected dyes displayed fluorescence emission in the “therapeutic window” and contained the carbaldehyde functional unit susceptible of passing on the affinity towards the bio-materials. Three dyes which were characterized by fluorescence emission in both solution and aggregated state were subjected to cytotoxicity determinations in order to verify their adequacy to medical applications. In line with this answer in the manuscript was introduced the following text in lines 218-220:
“…displaying fluorescence emission in solution above 650 nm and containing the carbaldehyde functional unit susceptible of passing on the affinity towards the bio-materials…”

Comments: What was the idea behind the synthesis of Zn-, Cu-, Ni- and Pd-complexes of the porphyrins?

Answer: The optical properties of porphyrin based absorbing chromophores can be modulated by changing the nature or position of the substituents as well as metal insertion. In our experiments we selected d-block elements possessing polarizable valence electrons capable of perturbing the π-electron system of the porphyrin core. In line with this answer in the manuscript was introduced the following text in lines 113-116:

“Metalation of halogenoMPP AB3 type 2a,b and trans-A2B2 type 3a,b with d-block elements possessing polarizable valence electrons capable of perturbing the π-electron system of the porphyrin core was performed in order to underscore the effect of metal core substitution on their reactivity and UV-Vis optical properties”.

Two other remarks:

  • Porphyrins 6 and 7 differ only in a halogen atom situated well afar from the porphyrin chromophore; yet, the absorption spectra of 6c/7c and 6d/7d pair demonstrate different patterns: one Q-band for 7c and 7d vs. two Q-bands for 6c and 6d. That looks strange. Could the authors explain that?

Answer: Literature data indicate that in the regular absorption spectra of metalloporphyrins one of the Q bands is the electronic origin of the lowest-energy excited singlet state, while the second band (of higher energy ) includes one mode of vibrational excitation [M. Roundhill, Photochemistry and Photophysics of Metal Porphyrins, Phthalocyanins and Metal Ions in Supramolecular Systems in Photochemistry and Photophysics of Metal Complexes © Springer Science+Business Media New York 1994, chapter 9, p 321.]. Porphyrins 6 and 7 differ not only in the halogen element (Br/Cl) but also in the position of substitution of the phenothiazine unit relative to the heterocyclic N atom: in 6 bromine substituent is attached in a para position offering a privileged interaction by electronic effects, as compared to 7 where the chlorine substituent is situated in a meta position prevented by this electronic interaction. We believe that these structural features may lead to differences in the vibrational modes of the halogen-MPP with repercussions on the absorption spectra of Ni(II) and Cu(II) complexes in the Q bands region.  

2) Fig. 2 does not look consistent with Table 3. According to table 3, the emission quantum yield for 9 is twice that for 14, while the emission quantum yield for 13 is only 20% larger that of 9. However, the emission intensity shown on Fig. 2 gives another impression of the relative quantum yields of 9, 13 and 14: the emission of 13 is at least two times stronger than that of 9. Are the emission spectra on Fig. 2 recorded for the same absorbance at the excitation wavelength? 

Answer: The emission spectra presented in table 3 and figure 2 were recorded after excitation at 420 nm. The difference is due to the variation in the concentration of the sample 13.

Reviewer 3 Report

The manuscript entitled "Novel phenothiazine bridged porphyrin-(hetero) aryl dyads: synthesis, optical properties, in vitro cytotoxicity and staining of human ovarian tumor cell lines." of Castelia Cristea et al., reports the synthesis and spectroscopic characterizations of new A4-type phenyl bridged porphyrin-phenothiazine dyads obtained by Suzuki-Miyaura cross coupling. I have already reviewed a paper by the same authors and they have resubmitted the manuscript after a drastic revision and addition of biological studies.

Despite the laudable effort and appreciable attempt to improve the interest in this article, my personal opinion on the work is to be reconsidered after major revision.

The authors must justify the presence of the phenothiazine group and correlate the structure with the chemical-physical properties. The authors report the results on the internalization studies obtained with only molecule 13 although they write to have tested also molecules 9 and 15. Explain why only molecule 13 shows a biological activity and correlate it with the other synthesized molecules. It is essential to highlight the particular reactivity of these specific molecules compared to similar studies (fluorescence microscopy and TPE-FLIM) on other porphyrins reported in literature. In addition, the internalization studies have been carried out using TPE-FLIM technique on molecules that are not NIR candidates, as incorrectly reported from the authors. Then the authors must correct the phrases indicating "NIR dyes candidates" in lines 34, 93, 206, 706.

Here are some typos in the text:

Line 64: insert “-“ between Suzuki and Miyaura

Line 85: correct Myiaura with Miyaura

Line 203: In table 3, the stokes shift data are incorrect.

Author Response

The authors are thankful to the reviewers for the detailed analysis signalizing the weaker parts of our manuscript. We found the reviewers comments very helpful for improving the quality of the manuscript.

The manuscript entitled "Novel phenothiazine bridged porphyrin-(hetero) aryl dyads: synthesis, optical properties, in vitro cytotoxicity and staining of human ovarian tumor cell lines." of Castelia Cristea et al., reports the synthesis and spectroscopic characterizations of new A4-type phenyl bridged porphyrin-phenothiazine dyads obtained by Suzuki-Miyaura cross coupling. I have already reviewed a paper by the same authors and they have resubmitted the manuscript after a drastic revision and addition of biological studies.

Despite the laudable effort and appreciable attempt to improve the interest in this article, my personal opinion on the work is to be reconsidered after major revision.

Comments: The authors must justify the presence of the phenothiazine group and correlate the structure with the chemical-physical properties.

Answer: a comparison between the fluorescence properties of the synthesized compounds was introduced in the text lines 199-214 as follows:

“The presence of the phenothiazine units in dyes scaffolds, may be considered as responsible on one hand for the energy dissipation by typical butterfly vibrations thus inducing low fluorescence emission in solution, but on the other hand, for enlarging the intermolecular distance due to its bent structure thus preventing the intermolecular π-π stacking and thereby stimulating fluorescence in aggregated state [27]. In solution, the parent MPP dyes displayed fluorescence emissions typical to the porphyrin core in 660-670 nm range [4], while the extension of the pending phenothiazine arms of A3B type and A2B2 type MPP with additional peripheral phenothiazine or phenyl units produced fluorescence quenching in compounds 8, 11. The presence of formyl auxochromic group in peripheral positions appeared beneficial by imparting molecular polarization by electron withdrawing effect and thus fluorescence emission was recorded for compounds 9, 12, 13, 15 as shown in table 3. Phenothiazine bridged porphyrin-carbazole dyads A3B type 10 and A2B2 type 14 displayed fluorescence emissions in similar range. The symmetrical A4 type phenylene bridged porphyrin-(hetero)aryl dyads did not display fluorescence emission regardless the nature of the peripheral units: formyl-phenothiazine 17, N-phenyl carbazole 18, or phenyl-carbaldehide 19.”

Comments:  The authors report the results on the internalization studies obtained with only molecule 13 although they write to have tested also molecules 9 and 15. Explain why only molecule 13 shows a biological activity and correlate it with the other synthesized molecules.

Answer: A correlation between the biological activity of the synthesized molecules was introduced in text lines 273-279 as follows:

“The typical chemical reactivity of the carbaldehyde unit was the selected structural feature meant for passing on the affinity towards the bio-materials. In phenothiazine bridged porphyrin-phenothiazine dyads A3B type 9 and A2B2 type 13 the formyl group was attached to the bent phenothiazine unit(s) prone to prevent the intermolecular π-π stacking and green fluorescence was displayed by these staining agents in aggregated state. Dye 15 containing the formyl group attached to peripheral flat phenylene units exhibited aggregation-caused fluorescence quenching which limited its application.”

Comments: It is essential to highlight the particular reactivity of these specific molecules compared to similar studies (fluorescence microscopy and TPE-FLIM) on other porphyrins reported in literature.

Answer: The following text containing a supplementary bibliographic reference was added in lines 264-271:

“Fluorescence microscopy became an indispensable tool in single-molecule investigations, providing a high signal-to-noise ratio for visualization while still retaining the key features in the physiological context of native biological systems. [30]. Fluorescent porphyrin derivatives characterized by large Stokes shift between absorption (~ 400 nm) and red/NIR emission were prospected in the fluorescence detection in various tissues or in vivo. In order to prevent the aggregation caused quenching effect due to π-π stacking aggregation favored by the planarity of the aromatic structure of the macrocycle, one chemical approach was to introduce various bulky groups as substituents of the porphyrin core [31]”.

Comments: In addition, the internalization studies have been carried out using TPE-FLIM technique on molecules that are not NIR candidates, as incorrectly reported from the authors. Then the authors must correct the phrases indicating "NIR dyes candidates" in lines 34, 93, 206, 706.

Answer: the phrase “NIR dyes candidates” was replaced by “fluorophores in red spectral region”

Comments: Here are some typos in the text:

Line 64: insert “-“ between Suzuki and Miyaura

Answer: the dash was inserted in Suzuki-Miyaura

Line 85: correct Myiaura with Miyaura

Answer: The “Myiaura” was corrected to “Miyaura”

Line 203: In table 3, the stokes shift data are incorrect.

Answer: The Stokes shift values were recalculated and the corrected values were introduced in Table 3.

Round 2

Reviewer 1 Report

The authors sincerely responded and revised to the comments.
Now, I think this manuscript is suitable for publication.

Author Response

We thank to the reviewer for the approval of our work.

Reviewer 2 Report

The authors addressed all my comments. I think the manuscript is suitable for publication.

Minor remarks:

  • p.5, line 156: "A UV.."
  • p. 6, Figure 1: the dimensions of ε should be M-1cm-1, like in Table 1

Author Response

We performed the corrections suggested by the reviewer as follows:

-p5 line 156: “An” was corrected to “A”

-p6 figure 1 was replaced by the corrected version where on the vertical axis “ε (mol-1cm-1)” was replaced by “ε (M-1cm-1)” as in Table 1

We thank to the reviewer for the approval of our work.

Reviewer 3 Report

The manuscript entitled "Novel phenothiazine bridged porphyrin-(hetero) aryl dyads: synthesis, optical properties, in vitro cytotoxicity and staining of human ovarian tumor cell lines." of Castelia Cristea et al., has been revised by the authors and important details have been included. The authors replied to almost all of my comments, but a further study on the TPE-FLIM spectroscopic technique would be appreciated, with reference to other porphyrins or similar molecules.

My personal opinion on the manuscript is to be accepted after minor revision.

Author Response

As suggested by the reviewer we introduced a section summarizing recent literature data on TPE-FLIM technique using porphyrin derivatives in the text lines 297-308 as follows:

“In recent literature, functional porphyrins were specified as potential fluorescent probes for TPE fluorescence imaging of various tumoral cells. Thus, 2-acetyl-6-dimethyl-aminonaphthalene–porphyrin dyads, characterized by intramolecular fluorescence resonance energy transfer process (FRET) from the aminonaphthalene donor to the porphyrin acceptor unit, were employed for TPE fluorescence imaging of lung cancer A549 cells under the irradiation with a 740 nm femtosecond laser. [33,34] Mesoporous silica nanocomposites containing the photosensitizer 5,10,15,20-tetrakis (1-methyl 4-pyridinio) porphyrin tetra (p-toluenesulfonate) were employed for targeted TPE-FLIM of human breast carcinoma cell lines MCF-7 and human lung cancer A549 cell lines [35], while non-covalent complex of lipid-coated semiconductor CdSe/ZnS quantum dots with partially hydrogenated porphyrin (chlorin e6) photosensitizer was imaged within plasma membrane and intracellular compartments of living HeLa cells [36].”

And consequently, in the References section citations 33-36 in lines 865-876:

  1. Zhu M., Su C, Xing P., Zhou Y., Gong L., Zhang J., Du H., Bian Y., Jiang J., An AceDAN–porphyrin(Zn) dyad for fluorescence imaging and photodynamic therapy via two photon excited FRET, Inorganic Chemistry Frontiers, 2018, 5, 3061-3066, DOI 10.1039/c8qi00705e.
  2. Zhou M., Xing P., Gong L., Su C., Qi D., Du H., Bian Y., Jiang J., Two-Photon Excited FRET Dyads for Lysosome-Targeted Imaging and Photodynamic Therapy, Inorganic Chemistry, 2018, 57 (18), 11537-11542 DOI: 10.1021/acs.inorgchem.8b01581.
  3. Li S., Zhang Y., He XW., Li WY., Yu-Kui, Zhang YK, Multifunctional mesoporous silica nanoplatform based on silicon nanoparticles for targeted two-photon-excited fluorescence imaging-guided chemo/photodynamic synergetic therapy in vitro, Talanta, 2020, 209, 120552-120560.
  4. Valanciunaite J., Klymchenko A.S., Skripka A., Richert L. Steponkiene S., Streckyte G., Melyc Y., Rotomskis R., Non-covalent complex of quantum dots and chlorin e6: efficient energy transfer and remarkable stability in living cells revealed by FLIM, RSC Advances, 2014, 4, 52270-52278, DOI: 10.1039/C4RA09998B